# Recent Advances and Future Directions in Alzheimer’s Disease Genetic Research

**DOI:** 10.3390/ijms26167819

**Published:** 2025-08-13

**Authors:** Mikaela Stancheva, Draga Toncheva, Sena Karachanak-Yankova

**Affiliations:** 1Department of Genetics, Faculty of Biology, Sofia University “St. Kliment Ohridski”, 1164 Sofia, Bulgaria; mikaelas@uni-sofia.bg; 2Bulgarian Academy of Sciences, 1000 Sofia, Bulgaria; dragatoncheva@gmail.com; 3Department of Medical Genetics, Medical Faculty, Medical University-Sofia, 1431 Sofia, Bulgaria

**Keywords:** early-onset Alzheimer’s disease, late-onset Alzheimer’s disease, rare pathogenic variants, common risk polymorphisms, cell-free brain-derived DNA

## Abstract

Alzheimer’s disease (AD) is a complex neurodegenerative condition which, despite its high prevalence and socioeconomic impact on the world, has an etiology that remains poorly understood. The genetic causes of AD are complex and have been continuously studied for decades. They range from rare pathogenic, highly penetrant mutations in early-onset (EOAD) forms, which account for 5% of the cases to multiple-risk alleles across different genes in late-onset (LOAD) forms. Monogenic causes of EOAD allocate within *APP*, *PSEN1*, and *PSEN2* genes in 10–15% of cases. The most significant risk factor in LOAD heritability is the *APOE* ε4 allele, as well as numerous loci within genes involved in immunity, endocytosis, lipid metabolism, and amyloid and tau processing. LOAD can also be attributed to the accumulation of somatic mutations, which may be detected by analysis of brain-derived cell-free DNA (cfDNA) in plasma. This review offers a comprehensive overview of the genetic architecture of Alzheimer’s disease, with particular focus on the molecular mechanisms underlying both early- and late-onset forms of the condition. An improved understanding of the genetic etiology of AD can aid better prevention, earlier diagnosis, and novel therapeutic approaches. This can be achieved by analyzing understudied populations, performing case-control studies with appropriately matched controls, and surveying brain-derived cell-free DNA in plasma, with the latter having the potential to contribute to the implementation of liquid biopsy in dementology.

## 1. Introduction

Dementia is an “umbrella” term, usually used to describe a range of neurological conditions closely related to chronic and progressive loss of cognitive skills to such an extent that it interferes with a person’s daily life and activities [1]. It goes beyond normal cognitive decline associated with aging and it is usually preceded and often accompanied by behavioral and psychological symptoms (changes in mood, behavior, motivation, and general emotional instability) [2]. As life expectancy increases and mortality rates in younger populations decline, the global incidence of dementia has risen. Dementia is the seventh leading cause of death and one of the leading causes of disability within the elderly population worldwide. Since almost every country has a statistically aging population, it is expected that the numbers of dementia cases will increase to 78 million by 2030 and the disturbing 139 million by 2050 [3]. Although in high-income nations age-specific dementia rates have decreased, likely due to improvements in education, nutrition, and lifestyle, the growing prevalence of diabetes and obesity-related conditions could potentially reverse this trend, as highlighted in various editions of the World Alzheimer Report over the years. Delaying the onset of Alzheimer’s disease by just 5 years can have a tremendous effect on the economic burden and is estimated to be able to reduce it by 36% by 2050 [4].

Dementia can result from a wide range of health conditions, including stroke and pre-existing neurological disorders. It may also arise as a consequence of certain infections, such as HIV [5], or from harmful behavioral patterns, particularly chronic alcohol or drug abuse. The misuse of medications that can impair cognitive function, including sleep aids, anxiolytics, analgesics, and anticholinergics, can contribute to the development of the condition as well [6]. Other possible causes include repeated brain trauma leading to chronic traumatic encephalopathy, hearing loss, some inflammatory processes, or even certain nutrient deficiencies, such as a lack of vitamin B12 [7].

Dementia can take multiple forms, with Alzheimer’s disease (AD) being the most common, contributing to between 60% and 70% of the total number of cases [8]. Other major forms of the condition include vascular dementia, Lewy body dementia, and frontotemporal dementia [9]. Alzheimer’s disease is a complex neurodegenerative condition [10] affecting approximately 5–10% of individuals over the age of 65, and nearly 50% of those aged 85 and older.

The rising prevalence and substantial socio-economic burden of Alzheimer’s disease underscore the urgent need for novel and integrated approaches for its prevention, early diagnosis, and treatment. In this context, this review aims to provide a thorough and comprehensive overview of the genetic architecture of Alzheimer’s disease, with a primary emphasis on molecular mechanisms underlying its pathogenesis. It synthesizes recent key discoveries that can shape the development of future diagnostic strategies and therapeutic approaches. In light of ongoing advances in the field, this review holds the potential to serve as a valuable resource for clinicians and researchers engaged in Alzheimer’s disease research and care.

## 2. Hypothesis of Alzheimer’s Disease Etiology

Despite its widespread prevalence and significant socioeconomic impact, the complex and multifaceted causes of Alzheimer’s disease are still not completely understood. There are two primary hypotheses regarding the cause of Alzheimer’s disease: the cholinergic and the amyloid hypothesis.

### 2.1. The Cholinergic Hypothesis

This is the earlier proposed hypothesis, which suggests that the disease results from reduced levels of the neurotransmitter acetylcholine. As widely recognized, acetylcholine is essential for neuronal transmission, contributing significantly to cerebral cortical development and overall cortical activity [11]. In Alzheimer’s disease, it is believed that the cholinergic neurons in the nucleus basalis of Meynert, a structure in the basal forebrain, decline in number, resulting in various symptoms, with memory loss being the most prominent [12]. In a healthy adult brain, the nucleus basalis contains approximately 500,000 cholinergic neurons, whereas in advanced Alzheimer’s disease, this number may decline to just 100,000, undergoing extensive degeneration of over 75% [10], as these neurons experience selective loss, resulting in cholinergic deficiency [13]. Cholinergic neurons in the basal forebrain have been found to exhibit tau pathology in both patients with mild cognitive impairment and those with Alzheimer’s disease [14]. The medial temporal lobe receives input from multiple neurotransmitter systems, including cholinergic innervation from the basal forebrain [15]. Biochemical and histologic research has shown a decline in cortical cholinergic activity and a loss of cholinergic basal forebrain neurons, leading to the development of the cholinergic hypothesis of Alzheimer’s disease [11].

### 2.2. The Amyloid Hypothesis

The amyloid hypothesis suggests that the buildup of amyloid plaques around and on neurons is the primary cause of Alzheimer’s disease and is based on the accumulation of the product of the amyloid precursor protein (*APP*) gene located on chromosome 21 [16]. This is supported by the observation that in patients with Down syndrome, having an extra copy of this gene nearly always leads to the appearance of amyloid plaques by the fourth decade of life and at least the initial symptoms of the condition in the fifth and sixth decade of life [17]. Autopsies of deceased patients diagnosed with Alzheimer’s disease provide practical evidence supporting these theories. Amyloid plaques are typically associated with Alzheimer’s disease, Down syndrome, or, to some extent, general aging [18]. The density of plaques, as well as their quantity and distribution, are generally positively associated with the severity of condition onset [19]. However, the exact molecular mechanisms remain unclear, as it is still uncertain whether this event is a consequence or a cause of the neuropathological process. This uncertainty has persisted since the initial description of the amyloid theory and the study of APP aggregation and misfolding [20].

Therefore, it is believed that Alzheimer’s disease may be caused by the buildup of abnormally high levels of amyloid-beta (Aβ) protein in the brain, which leads to the formation of amyloid plaques [21]. On the other hand, the buildup of tau proteins, forming neurofibrillary tangles, is also commonly observed as a universal sign of neuron death [22]. In Alzheimer’s disease, tau undergoes abnormal phosphorylation, causing it to detach from microtubules and form twisted tangles inside neurons, known as neurofibrillary tangles [23]. These tangles interfere with neuronal function and are one of the hallmark features of Alzheimer’s pathology, contributing to the cognitive decline seen in patients [24]. In the adult human brain, six tau isoforms are produced through the alternative splicing of a single tau gene located on chromosome 17 [25]. Three of the tau isoforms contain three tandem microtubule-binding repeats, while the remaining three have four [26]. Single-cell gene expression profiling has shown a change in the ratio of three-repeat to four-repeat tau in individual human cholinergic basal forebrain neurons within the nucleus basalis and CA1 hippocampal neurons as Alzheimer’s disease progresses, but not during normal aging [27]. Both amyloid plaques and neurofibrillary tangles contribute to physical blockages in neurons, making them generally unable to transmit impulses, which ultimately leads to a decline in brain function, particularly related to memory [10]. The ability to break down pathological proteins and prevent their accumulation is age-related and regulated by brain cholesterol [28].

Abnormal Aβ processing is not the sole mechanism contributing to Alzheimer’s disease, nor does it fully capture its complexity. Many of the genes containing AD risk variants are highly expressed in immune-related tissues, like the spleen, as well as cell types, including astrocytes and microglia, highlighting the crucial contribution of immune system dysfunction to AD pathology [29]. Various reactive astrocyte subtypes have been identified in regions affected by Aβ or tau pathology, likely reflecting their involvement in the neuroinflammatory processes characteristic of AD. These processes are mediated through the release of cytokines, chemokines, reactive oxygen species, and other inflammatory factors [30].

Within this framework, the neuroimmunomodulation hypothesis has been proposed, suggesting that Alzheimer’s disease arises due to glial cell response to damage signals, which initiate a neuroinflammatory reaction. This response leads to the dysregulation of protein kinases and phosphatases, ultimately promoting tau protein hyperphosphorylation and oligomerization. Tau oligomers and filaments, released following the process of neuronal apoptosis, can further reactivate microglial cells, perpetuating a detrimental molecular signaling cascade that contributes to the ongoing neurodegeneration in Alzheimer’s disease and other tauopathies [31].

The pathological tau and Aβ processing are supported by the immunomodulation hypothesis as well as the infectious hypothesis, which suggests that the presence of a pathogen triggers an inflammatory response, leading to the aggregation of tau and Aβ, which in turn exacerbates neuroinflammation and contributes to disease progression [32].

### 2.3. Genetic Etiology

From a genetic perspective, the etiology of Alzheimer’s disease varies between familial and sporadic cases, which is often related to the age of the condition’s onset. Early-onset Alzheimer’s disease is characterized by the onset of clinical symptoms before the age of 65. Approximately 35% to 60% of individuals with EOAD have a first-degree relative affected by the condition, and in about 10–15% of these familial EOAD cases, the disease follows an autosomal dominant pattern of inheritance. The strong genetic basis of EOAD cases is attributed to single-gene mutations or to an increased polygenic burden—an aggregate cumulative genetic risk stemming from the combined effects of numerous variants, each with a minor contribution [33]. In contrast, late-onset Alzheimer’s disease forms are mainly sporadic and considered to be multifactorial—caused by the interplay of genetic and environmental factors [34]. In this context, the development of polygenic risk scores (PRSs) which integrate multiple genetic variants into a single metric to estimate an individual’s overall genetic risk for a particular phenotype, demonstrates considerable potential for predicting the risk of developing Alzheimer’s disease. Furthermore, late-onset sporadic AD forms may also result from the accumulation of somatic mutations within the brain [35].

#### 2.3.1. Genetics of Early-Onset Alzheimer’s Disease

Early-onset AD accounts for about 5% of all cases. Compared to the late-onset form, EOAD has a more aggressive course with atypical presentation and diverse phenotypic manifestations. It can be familial, usually autosomal-dominant, and caused by rare, pathogenic, highly penetrant mutations in single genes [36]. Mutations that lead to the development of early-onset Alzheimer’s disease are found within the *APP* (amyloid precursor protein), *PSEN1* (presenilin 1), and *PSEN2* (presenilin 2) [37] genes, the products of which regulate the production of Aβ. These mutations account for the genetic cause of 10–15% of early-onset cases [38].

The amyloid precursor protein gene (*APP*) encodes a transmembrane protein that is proteolytically cleaved by two distinct pathways: nonamyloidogenic and amyloidogenic, the last referring to the formation of amyloid plaques. In the nonamyloidogenic processing pathway, APP is cleaved within the Aβ domain by α-secretase with the formation of soluble APPα, which plays a role in neurogenesis, synaptogenesis, and sequestering of metal ions. In the amyloidogenic pathway, APP undergoes proteolysis by β-secretase and subsequent cleavage by γ-secretase in diverse sites (Figure 1). This process generates Aβ peptides with different C termini, namely Aβ40, which is the most abundant; Aβ42, which is more prone to Aβ aggregation; and others. Aβ is the main component of extracellular plaques, both diffuse and neuritic, that accumulate throughout the medial temporal lobe and cortex in the brains of individuals with Alzheimer’s disease, progressing into the deep gray nuclei, brainstem, and eventually the cerebellum in the later stages of the neuropathology [11,39]. The *APP* gene consists of 17 exons in total and encodes several isoforms produced through the alternative splicing of exons 7 and 8 [40,41], three of which are relevant to Alzheimer’s disease as a neurodegenerative condition (isoforms 695, 751, and 770) and are expressed solely in the central nervous system [42,43]. Exons 16 and 17 encode the portion of the APP protein that forms the Aβ fragment [44]. More than 50 pathogenic mutations have been identified in the *APP* gene [45], the most frequent of which is the Val717Ile/London mutation. Mutations in the *APP* gene lead to an elevated Aβ42/Aβ40 ratio, along with an increase in the levels of total tau and phosphorylated tau in neurons [46]. This suggests that an increased genetic load of *APP*, also observed in cases of familial Alzheimer’s disease, may contribute to the condition, although not in all cases [47]. The amyloid precursor protein can be found in various tissues, including those in the central nervous system [48].

The *PSEN1* gene encodes a component of a protein complex known as γ-secretase. This protein serves as the proteolytic unit within the complex, carrying out its primary function. The γ-secretase complex is located in cell membranes, where it induces cleavages at specific sites in certain transmembrane proteins. Several variants of *PSEN1* have been identified in patients with early-onset Alzheimer’s disease [46], as most of them are SNPs causing synthesis of an abnormal protein presenilin 1. The defective product disrupts the proper functioning of the gamma-secretase complex, leading to improper processing of the APP protein. This, in turn, causes an overproduction of the longer and more toxic form of amyloid-beta, which contributes to the formation of amyloid plaques and the onset of Alzheimer’s disease symptoms [49]. Several hundred mutations have been identified in *PSEN1*, most of which are classified as pathogenic [45], including missense mutations, small insertions, deletions, and genomic deletions [38]. Mutations in *PSEN1* show complete penetrance and are causative for the most severe forms of Alzheimer’s disease with an onset as early as the third decade [50].

Presenilin 2 functions similarly to presenilin 1 and is also involved in processing proteins that transmit signals from the cell membrane to the nucleus. Once these signals reach the nucleus, they activate genes crucial for cell growth and differentiation. Together with other enzymes, presenilin 2 participates in the processing of amyloid precursor protein and the generating of amyloid-beta protein. The two most common *PSEN2* mutations are single nucleotide changes that disrupt the normal production of amyloid precursor protein, resulting in the overproduction of amyloid-beta peptide [51,52]. Missense mutations in the *PSEN2* gene may exhibit incomplete penetrance, with carriers experiencing an older age of disease onset compared to those with *PSEN1* mutations. These missense mutations result in altered proteins that disrupt the fusion of the γ-secretase complex, which in turn affects the processing of *APP* and leads to an increased *Aβ42/Aβ40* ratio [53].

The remaining cases of early-onset AD (without mutations in *APP*, *PSEN1*, and *PSEN2*) are caused by mutations in other genes, such as *GRN* (progranulin), *MAPT* (microtubule-associated protein tau), *TREM2* (triggering receptor expressed on myeloid cells-2), *NOTCH3* (neurogenic locus NOTCH3 homologous protein 3), and *CLU* (clusterin), or the causative mutation remains unidentified. It is presumed that the early-onset of Alzheimer’s disease may be due to enrichment of common risk variants associated with late-onset AD [54] (Figure 2).

#### 2.3.2. Genetics of Late-Onset Alzheimer’s Disease

Late-onset (mainly sporadic) Alzheimer’s disease is a complex, multifactorial condition, resulting from a complex interaction between genetic susceptibility and environmental influences. Late-onset Alzheimer’s disease has a strong genetic predisposition, with heritability ranging from 40–80% [34]. The genetic architecture of late-onset Alzheimer’s disease is polygenic in nature and has been thoroughly investigated using genome-wide association studies (GWAS) for more than a decade [55].

Among the identified genetic factors, the ε4 allele of the apolipoprotein E gene remains the most significant risk factor for late-onset Alzheimer’s disease. Beyond *APOE* ε4, genome-wide association studies have uncovered more than 70 additional risk loci associated with increased susceptibility to the disorder [56]. Extensive research on late-onset Alzheimer’s disease has demonstrated that genetic heritability is closely linked to pathways involving microglial function, neuroinflammation, amyloid precursor protein processing, lipid metabolism, tau protein dynamics, and endocytic mechanisms [39].

The *APOE* gene encodes apolipoprotein E which binds lipids to form lipoproteins and plays a vital role in lipid transport and metabolism. Among its three major isoforms, APOE3 (Cys112Arg158) is the most prevalent, APOE2 (Cys112Cys158) is associated with a reduced risk of Alzheimer’s disease [57], while APOE4 (Arg112Arg158) significantly increases disease susceptibility. These isoforms are determined by the combination of two coding polymorphisms rs429358T>C and rs7412T>C, as the defining haplotypes ε2, ε3, and ε4 determine the APOE2, APOE3, and APOE4 isoforms, respectively (Figure 3). The presence of one *APOE ε4* allele raises the risk of developing late-onset Alzheimer’s disease approximately three- to fourfold, whereas homozygosity for the ε4 allele confers a nine- to fifteenfold increase in risk [56]. The frequency of *APOE ε4* exceeds 40% among individuals diagnosed with Alzheimer’s disease but varies among populations—ranging from 9% to 20% in Caucasian, Japanese, African American, and Hispanic groups [56]. Furthermore, it was established that the association between Alzheimer’s disease risk and *APOE ε4* allele dosage is less significant in Hispanic and African American populations, yet more pronounced in Japanese individuals relative to Caucasians [56]. Additionally, the *APOE ε4* allele has been linked to an earlier age of disease onset. APOE competitively binds to Aβ receptors, such as LRP1 (low-density lipoprotein receptor-related protein 1), located on the surface of astrocytes, effectively blocking the uptake of Aβ. This results in impaired clearance of Aβ and its early stages of deposition. Some studies even suggest that APOE may “influence tau-mediated neurodegeneration and microglial responses to AD-related pathologies” [56]. The *APOE ε4* allele has been associated with herpes simplex virus type 1 infections [58]. This proves to be particularly significant, as it highlights the connection between infectious diseases and Alzheimer’s disease, presenting potential opportunities for both diagnostic advancements and therapeutic interventions. Emerging evidence suggests that even the ‘neutral’ APOE3 isoform may contribute to Alzheimer’s disease pathology by modulating tau protein dynamics, potentially affecting both the severity and progression of the disease [59]. These and related findings reinforce the view that *APOE* polymorphisms contribute to Alzheimer’s disease susceptibility by affecting both beta-amyloid deposition and tau protein processing [57].

The microtubule-associated protein tau—product of the *MAPT* gene—is primarily expressed in neurons and plays a key role in the assembly and stabilization of microtubules [60]. The dysfunction of tau protein, either through mutations or abnormal processing, accelerates the neurodegenerative process in Alzheimer’s disease. Studies have demonstrated that pathological tau propagates throughout the brain in a characteristic and sequential manner that aligns with the clinical progression of Alzheimer’s disease. This pattern of spread positions tau as a promising therapeutic target for interventions designed to slow or prevent disease advancement, underscoring the critical need to elucidate its role in neurodegenerative processes [61].

The bridging integrator 1 (*BIN1*) gene has been identified as LOAD related within the framework of the neuroimmunomodulation hypothesis and its role in membrane tubulation by interacting with microtubule-associated proteins like tau. Additionally, BIN1 is involved in key cellular processes, including intracellular beta-amyloid accumulation and early endosome enlargement [62,63]. Numerous other genes related to late-onset Alzheimer’s disease have been implicated in genetic susceptibility to the disease in support of the neuroimmunomodulation hypothesis. For example, the transcription factor *PU.1 (SPI1)*, which regulates the expression of immune-related genes in myeloid cells, may contribute to Alzheimer’s disease by modulating essential immune pathways and influencing the epigenetic landscape [64]. Additionally, the myeloid cell surface antigen *CD33*, which is expressed on microglial cells, plays a role in the negative regulation of cytokine production. As a result, it may influence Alzheimer’s disease by modulating microglial activation and the immune response in the brain [65]. The complement receptor 1—*CR1* gene, also known as *CD3S*, *C3b/C4b* receptor, belongs to the RCA (Regulators of Complement Activation) family. It encodes a monomeric type I membrane glycoprotein expressed on the surface of erythrocytes, leukocytes, and follicular dendritic cells in the spleen. In humans, *CR1* acts as a receptor for *C3b* and *C4b* and plays a crucial role in processing and clearing complement immune complexes. It also facilitates cell binding to particles labelled with an activated complement molecule [66]. The CR1 protein is involved in immune regulation, phagocytosis, and serves as an inhibitor of both classical and alternative complement pathways [67].

Clusterin, encoded by the *CLU* gene, is a type of apolipoprotein that plays a crucial role in several biological processes, including protein folding, lipid metabolism, and cellular apoptosis. Like APOE, it has chaperone properties and is abundant in the brain, including the hippocampus. It has been implicated in the pathogenesis of Alzheimer’s disease, in both early- and late-onset AD, through its involvement in amyloid-beta clearance from nerve tissue and cerebrospinal fluid (CSF), neuroinflammation, and cellular stress responses. Clusterin is involved in amyloid-beta clearance, and impaired function due to genetic variations may result in inefficient amyloid-beta removal, leading to plaque accumulation. Additionally, *CLU* modulates neuroinflammation by influencing microglial activation and plays a role in regulating complex formation, apoptosis, and lipid transport [68]. Although certain alleles may have protective roles under certain conditions by reducing oxidative stress and neuronal damage, *CLU* risk variants have been linked to exacerbated disease progression. In conclusion, clusterin’s dual role as both a protective and risk factor underscores the complexity of its involvement in AD and makes it a potential target for therapeutic strategies aimed at slowing or halting disease progression [69].

The *PICALM* gene was identified as the third most significant genetic susceptibility locus associated with Alzheimer’s disease after *APOE* and *BIN1.* It is located on chromosome 11 and has 24 splice variants, 15 of which encode proteins. Its product, the phosphatidylinositol-binding clathrin-assembly protein, is ubiquitously expressed in nervous tissues predominantly in micro vessels [70,71] and plays a role in the membrane repair of synaptic vesicles. It is found in both pre- and post-synaptic structures, where it functions in clathrin-mediated endocytosis. Mice with nonsense mutations in the *PICALM* gene exhibit abnormal hematopoiesis and iron metabolism, but do not show any noticeable neurological dysfunction [71]. The gene was first identified and connected to lymphoid and myeloid acute leukemia [23], but it was later identified in one of the first large-scale genome-wide association studies for LOAD [72] and further confirmed in subsequent GWAS [73].

The ATP-Binding Cassette Transporter 7 (*ABCA7*) gene was first identified in macrophages, as *ABCA7* is highly expressed in myeloid cells, particularly in granulocytes and monocytes [74]. Similarly to other members of the *ABC* family, the expression of *ABCA7* is influenced by changes in lipid concentration [75]. It has been implicated in Alzheimer’s disease due to its involvement in the clearance of amyloid-beta and the regulation of inflammation, two key processes associated with AD pathology. Studies have shown that mutations in *ABCA7* are associated with an increased risk of late-onset Alzheimer’s disease, with evidence suggesting that these mutations may impair the ability to clear amyloid-beta from the brain, contributing to amyloid plaque buildup [76]. Furthermore, *ABCA7* is involved in immune responses, and its altered function may also influence microglial activity, exacerbating neuroinflammation, which is another hallmark of Alzheimer’s [77].

Other genes found to be associated with an increased risk of Alzheimer’s disease in a large genome-wide study are those coding three proteins from the *MS4A* family (membrane-spanning 4-domains), namely *MS4A4A*, *MS4A4E*, and *MS4A6E* [78]. Most genes in this family encode proteins that feature four or more transmembrane domains, along with cytoplasmic domains at both the amino and carboxyl termini, which are typically encoded by separate exons. Although the precise role of *MS4A4A*, *MS4A4E*, and *MS4A6E* in Alzheimer’s disease is still under investigation, current evidence suggests that genetic variations in these proteins may affect immune function and microglial activity in the brain—key elements in the development of Alzheimer’s [79]. Dysfunction in these MS4A proteins may disrupt this clearance process, leading to amyloid plaque buildup and increased neuroinflammation, which in turn accelerates neurodegeneration. As such, the MS4A family proteins offer promising avenues for further research into the genetic underpinnings of Alzheimer’s disease and may open doors for future therapeutic interventions [55].

The *sorLA* gene is predominantly expressed in the central nervous system and encodes a type 1 transmembrane protein composed of 2214 amino acids, known as the sortilin-related receptor. This receptor plays a key role in the endo-lysosomal pathway by binding specific peptides and membrane-bound proteins and directing them to the retromer multiprotein complex for proper sorting. Disruptions in this pathway due to *sorLA*-related retromer dysfunction lead to endosomal traffic issues, one of the earliest cellular abnormalities observed in both familial and sporadic Alzheimer’s disease. Studies of brains of Alzheimer’s disease patients have shown decreased *sorLA* expression [80] as well as lower retromer protein subunit levels in the transentorhinal cortex [81]. The *sorLA*-*VPS26B* retromer complex has also been linked to the regulation of amyloid precursor protein [82]. Thus, in vitro studies support the idea that elevated *sorLA* activity is associated with more efficient APP processing and decreased Aβ production, whereas loss of *sorLA* function commonly results in increased APP accumulation, thereby promoting the formation of amyloid plaques [83].

Additionally, the pathological processing of tau and Aβ, along with other neuropathological events, may be influenced by genes associated with metabolic disorders, particularly those involved in lipid metabolism, atherosclerosis pathways, and insulin resistance [84]. In this regard, the solute carrier family 10 member 2, *SLC10A2*, plays a crucial role in cholesterol metabolism regulation. Studies have shown that excessive cholesterol accumulation in neurons can result in neuronal death, memory impairment, and heightened Aβ production [69]. Given that AD may be viewed as a metabolic disorder partially driven by insulin resistance [84], it is suggested that the zinc finger CW-type PWWP domain protein 1, *ZCWPW1*, which plays a role in the positive regulation of DNA metabolic processes, may reduce the risk of LOAD by mitigating insulin resistance [85]. Additionally, genes involved in the AMP-activated protein kinase (AMPK) pathway have been linked to Alzheimer’s disease risk through their role in regulating energy balance, as well as glucose and lipid metabolism [86], autophagy dysfunctions leading to Aβ and tau pathology [84], and alteration of the synaptic plasticity of hippocampal neurons [87]. Similarly, methylenetetrahydrofolate reductase, *MTHFR*, a key enzyme in the methylation cycle, has been linked to Alzheimer’s disease due to elevated levels of homocysteine found in affected individuals. High homocysteine levels are associated with vascular damage, increased inflammation, and dysfunction of endothelial cells [88]. Additional research is necessary to investigate the genetic links between chronic degenerative diseases, like diabetes mellitus, and Alzheimer’s disease, as well as to uncover the neuropathological pathways involved.

Other genes involved in LOAD molecular pathogenesis are *TREM2*, a triggering receptor expressed on myeloid cells 2, linked to tau levels in cerebrospinal fluid [89], as well as the amyloid-hypothesis-suggested disintegrin and metalloproteinase domain-containing protein 10 (*ADAM10*), which play a role in the pathological processing of amyloid-beta. *ADAM10* is the primary α-secretase in the brain and is a potential AD biomarker [74], showing altered levels in platelets, plasma, serum, and CSF in AD patients [90].

The genes implicated in early- and late-onset AD development, their prevailing mutation type according to Human Genome Mutation Database, and the mechanism of their involvement in AD pathology are summarized in Table 1; as GWAS detected variants in the LOAD genes, their gnomAD frequency and odds ratio are given in Appendix A.

The critical role of gene–environment interactions in the multifactorial etiology of Alzheimer’s disease can be mediated by epigenetic mechanisms. The timing of environmental risk factors for AD appears to be pivotal, with early-life and developmental periods representing windows of heightened epigenetic sensitivity that may shape lifelong AD risk [91]. Genome-wide and gene-specific studies have revealed differential DNA methylation patterns in AD brains and peripheral tissues—such as hypomethylation at *APOE* and *TREM2* loci, hypermethylation of *BDNF* (brain-derived neurotrophic factor) and *SPINT1* (Serine Peptidase Inhibitor, Kunitz Type 1) promoters, and global methylation shifts linked to *APOE ε4* status that correlate with disease progression and cognitive decline, suggesting their utility as diagnostic or prognostic biomarkers [76]. At the chromatin level, AD-related histone acetylation and deacetylation imbalances—mediated by enzymes like histone deacetylases 2 and 6 and sirtuins—have been shown to impair synaptic plasticity, memory formation, and microglial function in both postmortem tissue and animal models, with histone deacetylases inhibitors demonstrating potential cognitive benefits. Furthermore, dysregulation of multiple non-coding RNA classes—including miRNAs (e.g., miR-29, miR-206, miR-200b/c), long non-coding RNAs (e.g., *BACE1-AS*, *51A*, *EBF3-AS*), circular RNAs, and alterations in small nuclear RNA aggregates—has been implicated in APP processing, amyloid accumulation, tau pathology, and neuroinflammation [92].

As in many other polygenic disorders, most of the studies of their heritability are performed on Caucasian populations, as recent genetic studies of Alzheimer’s disease in non-European populations have revealed both shared and population-specific risk factors [93], highlighting the importance of conducting further studies across diverse populations to identify ethnic-specific polymorphisms [86]. For instance, African Americans are twice as likely to develop AD [55], emphasizing the need for more studies involving these populations. In Latin American cohorts, particularly in Mexico and Colombia, autosomal-dominant *PSEN1* mutations such as A431E and E280A have been identified as major contributors to early-onset familial AD. Notably, protective variants like *APOE3-Christchurch* and a rare *RELN* (reelin) variant have been observed in Colombian individuals carrying the *PSEN1* E280A mutation, significantly delaying disease onset and suggesting genetic resilience mechanisms [93]. In South America, the first genome-wide association study conducted in Argentina and Chile, involving admixed populations, confirmed the known *APOE ε4* risk allele and identified novel loci associated with lysosomal and immune function pathways, including *TREM2L*, *IGH*, *ADAM17*, and *PLCG2* [94]. The predictive power of European-derived polygenic risk scores was found to decline with increasing Native American ancestry, emphasizing the need for ancestry-informed risk modeling. Similarly, GWAS in East Asian populations (Chinese, Korean, and Japanese) have uncovered 26 AD-associated loci, with rare variants such as *TREM2* p.H157Y and *SHARPIN* p.G186R/p.R274W showing significant associations unique to these populations [95].

Women are disproportionately affected by AD, exhibiting higher incidence and more rapid cognitive decline compared to men [96,97]. Biological factors, including hormonal differences such as estrogen decline during menopause, are thought to contribute to this disparity by influencing amyloid-beta accumulation and tau pathology [98]. Moreover, genetic studies highlight sex-specific effects of the *APOE ε4* allele, with female carriers showing greater risk and earlier onset than their male counterparts [99], but still gender-specific genetic factors in AD are still understudied [100]. Some GWAS may be biased due to imbalances in sample sizes between controls and AD patients, as well as unaccounted-for factors such as comorbidities, gender, age range, environmental exposures, and medication use, which can influence the study results.

#### 2.3.3. Somatic Mutations in Alzheimer’s Disease

In addition to germline mutations, Alzheimer’s disease may also be due to the accumulation of brain somatic mutations in the course of aging [85,86]. This process releases DNA that can cross the blood–brain barrier and enter the blood plasma as brain-derived cell-free DNA (cfDNA). Several recent studies have highlighted the potential of neuron-derived cfDNA in blood plasma as an early diagnostic marker for neurodegenerative conditions [101]. By analyzing the methylation pattern differences between cortical neurons and plasma samples, researchers were able to detect neuron-derived cfDNA with high precision, successfully identifying all individuals with Alzheimer’s disease and distinguishing those with mild cognitive impairment who later progressed to AD. Furthermore, when cfDNA fragmentation patterns in plasma samples from individuals with Alzheimer’s disease and healthy controls were compared, the overall concentration of cfDNA was comparable between both groups, but notable differences in fragmentation patterns were detected, suggesting that these specific variations could be useful as diagnostic indicators for Alzheimer’s disease [102].

### 2.4. Microbiome

Emerging research highlights gut microbiota as a genetically influenced system that significantly impacts host physiology through production of metabolites such as short-chain fatty acids, peptides, and neurotransmitters [103,104]. The gut–brain axis, a bidirectional communication network that integrates neural and humoral signals, plays a key role in both neurological and gastrointestinal functions [105]. Experimental models, such as germ-free mice and antibiotic-induced dysbiosis, have demonstrated disruptions in the production of key neuroactive metabolites, including BDNF, gamma-aminobutyric acid, N-methyl-D-aspartate (NMDA) receptors, and tryptophan [106,107].

Gut dysbiosis may promote both local and systemic inflammation, compromising the intestinal epithelial barrier and increasing blood–brain barrier permeability, which leads to neuroinflammation and neuronal dysfunction [108]. Elevated levels of bacterial endotoxins like lipopolysaccharides (LPS) have been found in brains of Alzheimer’s disease patients, where LPS is found to co-localize with β-amyloid plaques, to enhance Aβ fibrillation, and to activate inflammatory signaling pathways such as NF-κB, thereby contributing to nerve inflammation [109,110]. Additionally, bacterial amyloid peptides from families like *Akkermansiaceae* and *Enterobacteriaceae* may facilitate Aβ aggregation, while beneficial microbes such as *Bifidobacteriaceae* have been found to be reduced in AD cases, highlighting the complex role of microbiome balance in disease progression of the condition [111].

Although the gut microbiome’s role in the processes of neurodegeneration has been extensively studied, the oral microbiome, which is directly exposed to pathogens and systemic signals, has received comparatively less attention. Nonetheless, emerging evidence links the oral pathogen *Porphyromonas gingivalis* and its virulence factors to increased AD risk via promotion of amyloid accumulation and neuroinflammation [112].

The oral–brain and oral–gut–brain axes facilitate constant communication via neural, immune, and endocrine pathways, where dysbiosis can trigger systemic inflammation and contribute to neurodegeneration and related systemic diseases that exacerbate cognitive decline [113]. Host genetics both shape and respond to these microbial shifts. For example, *APOE ε4* carriers exhibit altered oral and gut microbiota profiles, characterized by reduced microbial diversity and increased abundance of pro-inflammatory species. Moreover, polymorphisms in *TLR4*, a key innate immune system receptor, influence individual responses to LPS from oral pathogens, thereby modulating the severity of neuroinflammatory responses. Environmental exposures, including diet, pH, medication intake, smoking, alcohol, and oral hygiene, further interact with host genotypes to affect microbial composition and cognitive outcomes [113].

### 2.5. Metabolomics

The majority of metabolomic studies regarding neurodegenerative disorders focus on blood serum and cerebrospinal fluid, given their content of central nervous system biomarkers such as β-amyloid and tau proteins. One significant study proposes that other brain-specific metabolic dysregulations associated with early AD pathophysiology are systemically reflected in the peripheral blood serum; therefore, these metabolomic alterations hold promise as predictive biomarkers, offering a non-invasive approach for prognosis and stratification of individuals at risk for disease onset or progression [114].

## 3. Stages and Symptoms

The National Institute on Aging and the Alzheimer’s Association have created recommendations for the diagnosis and characterization of Alzheimer’s disease, highlighting three main stages of the condition: mild, moderate, and severe [115]. The mild stage of AD or mild cognitive impairment, often considered a transitional stage between normal cognitive aging and dementia, involves forgetfulness of recent information, loss of sense of time, getting lost in familiar locations, etc. The moderate stage of the condition usually involves forgetfulness of recent events and names, confusion while at home, worsening communication, need for help with personal care, personality changes, like repeated questioning, and more. The severe stage of AD is described by difficulty recognizing loved ones and relatives, general disregard for personal care, trouble walking, and deepening changes in character, frequently leading to aggressive outbursts [116].

Alzheimer’s disease usually has a long preclinical phase of up to 20 years with the pathophysiological process itself often beginning more than a decade before the clinical stage, and an average clinical duration of 8–10 years [117]. It is still unclear how long a patient can remain in any stage of the condition. The preclinical phase of the condition is followed by the prodromal stage, characterized by mild cognitive impairment with decline in one or more cognitive domains, observable both subjectively and objectively [118].

Alzheimer’s disease is very commonly accompanied by neuropsychiatric symptoms with depression and apathy being most common and present in more than 70% of major cognitive decline cases [119]. These cognitive manifestations, although common on their own or due to other conditions, do not have effective treatment options, although low doses of antidepressants, like a selective serotonin-reuptake inhibitor (e.g., escitalopram) can be applied [120]. The other common symptoms such as agitation, aggression, and psychosis have also been deemed as difficult to manage, as conventional and several “atypical” antipsychotics should be avoided due to serious risks of complications and death [121,122]. Additionally, patients diagnosed with AD that might be considered dangerous to themselves and/or others are usually hospitalized or institutionalized but rarely tested for additional psychological conditions. For instance, delirium in hospitalized and institutionalized patients most often goes undiagnosed [123] and therefore untreated, which can severely impede treatment, as antipsychotics are ineffective in delirium.

The efficient and accurate early diagnosis and staging of Alzheimer’s disease will benefit greatly from further implementation of the currently developing machine-learning frameworks, which can utilize region-specific structural MRI (magnetic resonance imaging) patterns as well as other diagnostic tool data [124].

## 4. Diagnostics

The most widely adopted diagnostic criteria for Alzheimer’s disease, both currently and historically, include those established by the National Institute of Neurological and Communicative Disorders and Stroke–Alzheimer’s Disease and Related Disorders Association (NINCDS-ADRDA) [125], the Diagnostic and Statistical Manual of Mental Disorders, Fifth Edition (DSM-5) [126] and the International Classification of Diseases, 11th Revision (ICD-11) [125].

Diagnosis of Alzheimer’s disease generally requires a comprehensive, multidisciplinary approach that integrates clinical assessment, cognitive evaluation, biomarker analysis, and neuroimaging techniques. The diagnostic process typically begins with a detailed review of the patient’s medical and family history and any comorbid medical conditions that may contribute to cognitive decline [127]. The initial assessment is typically followed by a thorough physical and neurological examination to detect motor, sensory, or other neurological signs that could assist in distinguishing Alzheimer’s disease from other potential diagnoses [125].

### 4.1. Neurobehavioral and Socio-Cognitive Assessment

Cognitive and neuropsychological assessments play a pivotal role in the diagnosis of Alzheimer’s disease, as they help detect impairments across multiple cognitive domains such as memory, executive functioning, attention, language, and visuospatial abilities. Commonly used tools for quantifying these deficits include the Mini-Mental State Examination (MMSE) and the Montreal Cognitive Assessment (MoCA) [128]. There are also multiple country-specific assessment tools, like the Linus Health Core Cognitive Evaluation and the Digital Clock and Recall DCAs by Linus Health, Inc. [129], as well as options for questionnaires in the native language of specific patients. For instance, in Japan the Wisconsin Card Sorting Test and the Trail Making Test, as well as the Clock Drawing Test (CDT) and the Cube Copying Test (CCT), are commonly used for screening, whereas the Neuropsychiatric Inventory, the Self-Rating Depression Scale, and the Geriatric Depression Scale are used occasionally [130].

Additionally, quality of life assessments can also be employed for the neurobehavioral and cognitive assessment of Alzheimer’s disease. The Alzheimer’s Society of the United Kingdom has pinpointed several exemplary questionnaires or instruments that can be used to measure the quality of life of people living with the condition, such as DEMQOL [131], QoL-AD (Quality of Life in Alzheimer’s Disease) [132], and DQoL (Dementia Quality of Life) [133].

Artificial intelligence can also have an enormous impact on AI diagnosis and management by enhancing early detection, risk prediction, and disease monitoring through advanced data analytics [134]. Machine learning models can analyze vast datasets, including MRI scans, PET images, genetic profiles, and cognitive assessments, to identify subtle biomarkers and predict disease progression with high accuracy. For instance, AI-based tools have shown promise in differentiating between mild cognitive impairment and early AD stages by detecting regional brain atrophy patterns from neuroimaging data [135]. Deep learning algorithms also aid in automating the segmentation of brain regions and quantifying amyloid or tau deposition in PET scans [136]. Furthermore, AI can integrate multi-omics data—genomic, transcriptomic, and metabolomic—to uncover novel diagnostic biomarkers and therapeutic targets [137]. Personalized medicine is becoming more feasible as AI helps stratify patients based on genetic and phenotypic risk factors, improving clinical trial design and individual treatment plans [138]. With continued advances, AI is poised to become an indispensable component in the clinical management and research of Alzheimer’s disease.

In summary, neuropsychiatric evaluation is essential due to the frequent occurrence of behavioral and psychological symptoms in individuals with AD, such as depression, apathy, anxiety, and agitation.

### 4.2. Laboratory Markers and Imaging Tools

Different laboratory tests, such as blood analyses and cerebrospinal fluid examination, are utilized to rule out alternative causes of cognitive impairment and to provide additional diagnostic support for Alzheimer’s disease. Key CSF biomarkers—namely, reduced concentrations of Aβ42 alongside increased levels of total tau and phosphorylated tau—have shown strong sensitivity and specificity in identifying Alzheimer’s pathology, making them valuable tools in both clinical practice and research contexts [139,140].

Neuroimaging plays a crucial complementary role in the diagnostic assessment of Alzheimer’s disease. Structural imaging techniques, such as MRI and computed tomography (CT), facilitate the visualization and therefore the detection of brain atrophy patterns indicative of AD, especially within the medial temporal lobe [141]. Functional imaging techniques, including fluorodeoxyglucose positron emission tomography (FDG-PET), can identify regions of decreased metabolic activity, most notably in the temporoparietal cortex [68]. Beyond structural imaging, molecular techniques like amyloid and tau PET scans can be utilized to detect hallmark pathological proteins of Alzheimer’s disease in vivo. The integrated use of structural, functional, and molecular imaging enhances diagnostic precision, proving particularly valuable during the early, preclinical, and prodromal stages of the disease.

### 4.3. Current and Future Digital Approaches

The integration of digital technologies with genomic research has opened new avenues for the early diagnosis and risk stratification of Alzheimer’s disease. In the context of human genetics, several digital approaches have already been deployed, while others are under development and show promise for future clinical translation. Current digital approaches include polygenic risk scores, next-generation sequencing with AI-based interpretation, large-scale digital biobanks (e.g., UK Biobank, Alzheimer’s Disease Neuroimaging Initiative) [142], and integrated genomic platforms as well as many more. Early and accurate diagnosis of Alzheimer’s disease is essential for better treatment results, and machine learning models are being designed to refine diagnostic accuracy by analyzing diverse data types [143]. Future digital approaches include digital twin modeling, i.e., generating dynamic computational replica of an individual, constructed using multi-omics, clinical, and environmental data, predictive genomics, CRISPR screening platforms with AI integration, etc. [144].

### 4.4. Molecular Diagnostic Tools

Circular RNAs (circRNAs), a unique class of non-coding RNAs characterized by their covalently closed-loop structures, have emerged as promising candidates with potential roles in both the prevention and treatment of AD [145] due to their remarkable stability, tissue-specific expression, and regulatory functions at the post-transcriptional level. Unlike linear RNAs, circRNAs resist exonuclease degradation, allowing them to persist in cells and biofluids, making them attractive biomarkers and potential therapeutic targets. In the context of AD, several circRNAs have been found to modulate key disease pathways, including amyloid precursor protein processing, tau phosphorylation, and neuroinflammation [146]. Notably, Leidinger et al., 2013 identified circulating miRNAs capable of distinguishing Alzheimer’s disease patients from healthy controls with accuracy above 90%, and from individuals with other neuropsychiatric conditions (schizophrenia, depression, bipolar disorder)—with ~76% accuracy [147]. Emerging evidence also suggests that dysregulated circRNAs may participate in ceRNA (competing endogenous RNA) networks, thereby influencing the expression of multiple AD-relevant genes through miRNA sequestration [148].

## 5. Therapy

### 5.1. Current and Potential Treatment

At present, a limited number of drugs for Alzheimer’s disease treatment have been approved, as they are aimed primarily at symptom management and slowing cognitive decline [125]. These include cholinesterase inhibitors (donepezil, rivastigmine, and galantamine) [125,149], monoclonal antibodies, such as Aducanumab, Lecanemab, Donanemab, as well as the NMDA receptor antagonist memantine, which modulates glutamatergic signaling [125], valproic acid, and rosiglitazone [150]. A recent advance in AD treatment is the novel rivastigmine nasal spray, which is used to treat mild and moderate forms of the disease. The intranasal delivery method aims to improve the bioavailability and targeting of therapeutic compounds to the brain [151,152].

Various immune system-modulating approaches are being explored for their ability to regulate neuroinflammation, promote neuronal survival, and slow disease progression. They focus on targeting specific immune receptors, like TREM2 [153], or enhancing microglial function through other immune receptors [154]. Additionally, cytokine-based therapies aimed at modulating pro-inflammatory and anti-inflammatory signaling pathways have also been explored [155]. Drugs, such as sargramostim (Leukine), are being explored for their potential to stimulate the immune system and reduce neuroinflammation in AD patients [156]. Research on fenamates, a type of non-steroidal anti-inflammatory drugs, has demonstrated their capacity to reduce inflammation and enhance memory performance in animal models, underscoring the promise of immune-targeted therapies for Alzheimer’s disease and related neurodegenerative conditions [157]. Moreover, immunotherapies such as daratumumab, which targets CD38, demonstrate potential for modulating the immune system in AD [158]. Bapineuzumab, a monoclonal antibody targeting amyloid beta promotes its removal from the brain while also modulating immune activity [159].

Nanobodies [160] and Fab fragments derived from full-length antibodies [161] possess unique properties such as small size and high antigen-binding specificity, making them valuable for precision therapy and diagnostic applications in AD [162]. Their reduced size [160] improves tissue penetration and lowers immunogenicity, enabling targeted drug delivery to specific cells and tissues [163]. For instance, a nanobody–drug conjugate designed to bind specifically to Aβ plaques and deliver anti-Aβ drugs holds significant potential as a targeted treatment for AD [164].

Additionally, boosting neurotrophic factors like BDNF and nerve growth factor (NGF), alongside immune-modulating agents, offers a comprehensive approach to treating neurodegenerative diseases. BDNF has proven more effective than NGF in restoring neural circuits, reducing neuronal loss, and improving brain function in Alzheimer’s disease [165].

Given that pro-inflammatory cytokines drive neuroinflammation in AD, targeting these molecules and their receptors offers therapeutic potential, with Traditional Chinese Medicine herbs like *Radix salviae*, ginseng, and licorice modulating key AD-related genes and pathways, demonstrating neuroprotective effects by reducing amyloid toxicity, neuroinflammation, and cognitive decline [166]. The JAK/STAT signaling pathway and Th17/Treg cell balance have been implicated in AD neuroinflammation, with compounds such as ganoderic acid A and ginsenosides showing promise in mitigating these effects [167].

In addition to pharmacological interventions, lifestyle modifications and dietary approaches have gained increasing attention as potential strategies to delay or mitigate the progression of Alzheimer’s disease [168]. The 2023 World Alzheimer Report highlights various examples in the search for “miracle brain food.” Numerous studies have shown that flavanols, found in berries, tea, wine, and certain vegetables, offer protective effects against neurodegenerative diseases [169]. Dietary patterns such as the Mediterranean diet, rich in fruits, vegetables, whole grains, fish, and healthy fats like olive oil, have been associated with a lower incidence of AD and slower cognitive deterioration [170]. The MIND diet, which combines elements of the Mediterranean and DASH (Dietary Approaches to Stop Hypertension) diets, emphasizes foods believed to promote neuroprotection, such as leafy greens, berries, nuts, and legumes, and has shown promising results in reducing AD risk [171]. Nutritional components like omega-3 fatty acids, antioxidants (vitamins C and E), and polyphenols have demonstrated neuroprotective effects in preclinical models by reducing oxidative stress and inflammation, both of which contribute to AD pathology [172]. When used alone or together with selegiline, vitamin E may also slow the progression of moderate Alzheimer’s disease [173]. However, clinical trials of single-nutrient supplements have yielded mixed results, underscoring the importance of a holistic dietary approach rather than isolated supplementation [172]. Moreover, lifestyle factors such as adequate sleep, stress management, and avoidance of smoking and excessive alcohol consumption are recognized as modifiable risk factors that influence cognitive health and may impact the onset and progression of AD [172]. Epidemiological studies suggest that cognitive engagement and social interaction may also reduce the risk of cognitive decline and improve brain health [168]. While these interventions are not curative, they represent accessible, low-risk strategies that could complement pharmacological therapies to improve outcomes for individuals at risk for or living with AD.

### 5.2. Future Treatment Directions

#### 5.2.1. Gene Therapy

Aducanumab has demonstrated the ability for blood–brain barrier penetration and for the promotion of Aβ clearance, showing significant clinical benefits in trials. However, other monoclonal antibody therapies like Bapineuzumab and Solanezumab have encountered setbacks, underscoring the need for more precise and patient-specific AD treatments [174], including gene therapy. Various gene therapy approaches employing recombinant adeno-associated virus (AAV) vectors expressing Aβ peptides or antibodies have shown to reduce the Aβ burden and plaque formation in transgenic mouse models, although cognitive effects were sometimes not assessed in these studies [175]. Additionally, delivery of enzymes like endothelin-converting enzyme or neprilysin via AAV vectors has been reported to decrease amyloid deposition, yet cognitive outcomes, once again, remain largely unexplored [176]. Overexpression of *TREM2* is found to improve microglial phagocytosis of Aβ, to reduce neuroinflammation, and to improve cognitive deficits in AD models, highlighting it as a promising therapeutic candidate [177]. Furthermore, short Aβ peptides (Aβ36–40) exhibit lower toxicity and can counteract Aβ42 toxicity. Alongside the intramuscular delivery by an AAV vector, the p75NTR ectodomain (physiological protective factor against amyloid-beta) has shown potential in mitigating AD pathology and cognitive impairments, though further research is needed [178].

Still, much of the ongoing research remains primarily focused on tau-targeting therapies [179]. A randomized clinical trial has demonstrated that an investigational antisense oligonucleotide BIIB080, also known as MAPT Rx, is generally well tolerated and is associated with a dose-dependent reduction in cerebrospinal fluid total tau and phosphorylated tau levels [180]. Furthermore, PET imaging revealed reduced tau accumulation in the assessed brain regions. BIIB080 is currently being evaluated in the Phase 2 CELIA study for early-stage Alzheimer’s disease.

Moreover, approaches like CRISPR-dCas9-mediated astrocyte-to-neuron conversion [181] and AAV-mediated delivery of anti-tau single-chain antibodies (scFvs) [182] have shown promise in reducing pathological tau accumulation in tauopathy models [183].

Gene therapy approaches utilizing AAV vectors explored in recent years include delivery of functional small molecules such as *R33*, aiming to stabilize the retromer complex, as well as RNA-based approaches with antisense oligonucleotides modulating gene expression [184]. In addition to gene therapy, several promising drug candidates, including Remternetug, an anti-amyloid antibody developed as a successor to Donanemab [125]; Buntanetap, an oral medication designed to promote the production of neuroprotective proteins with clinical trials for both Alzheimer’s and Parkinson’s diseases [185]; and Semaglutide, a glucagon-like peptide-1 receptor agonist, are currently under investigation as Alzheimer’s disease therapies.

#### 5.2.2. Gene-Editing Technology

CRISPR/Cas9 technology, pioneered by Emmanuelle Charpentier and Jennifer Doudna, has revolutionized gene editing and holds significant promise for advancing Alzheimer’s disease research by enabling the development of precise disease models and targeted therapeutic approaches [186]. In particular, base editing approaches, like the APOE4→APOE3r conversion, offer a precise and, potentially, safer alternative to traditional CRISPR-Cas9 editing for the correction of neurodegenerative disease-associated mutations [187].

Recent advances include the generation of a tau knockout mouse (tauΔex1) line using CRISPR/Cas9 targeting the *MAPT* gene, which overcame the limitations of earlier embryonic stem cell-based models and exhibited reduced seizure susceptibility without cognitive impairment [188]. The study suggests that precise manipulation of *SPl1/PU.1* expression results in a delayed AD onset [189]. These findings underscore the therapeutic potential of gene expression manipulation via CRISPR/Cas9 or other epigenetic tools to modulate immune and synaptic regulatory networks and may thus open new paths for Alzheimer’s disease treatment.

#### 5.2.3. mRNA Vaccine

Messenger RNA therapy presents a cutting-edge biomedical approach of delivering genetic instructions for the in-situ production of therapeutic proteins directly to the cell. While acute neurological conditions like stroke often necessitate immediate invasive interventions, chronic neurodegenerative diseases and brain tumors progress more gradually and are significantly complicated by challenges like the blood–brain barrier and multifactorial pathology. These characteristics make them particularly suitable and promising candidates for mRNA-based therapies. Ongoing research suggests that mRNA-based therapies hold the potential to revolutionize and transform both the prevention and treatment of such conditions [7,161,190].

Driven by the urgent need for more effective Alzheimer’s disease treatments and ongoing advancements in understanding its pathogenesis, numerous clinical trials have been initiated, the most cited of which are presented in Appendix A.

## 6. Conclusions

Alzheimer’s disease remains a major global health challenge, and despite years of research, effective treatments that modify the disease’s progression are insufficient. The extended prodromal phase of AD, coupled with the condition’s intricate nature, highlight the critical need for effective early detection strategies and innovative treatment approaches that tackle fundamental mechanisms of disease progression and not just surface symptoms [191,192].

In this context, the current review examines the recent advances and future directions in Alzheimer’s disease genetic research, emphasizing the monogenic determinants of early-onset Alzheimer’s disease as well as hereditary predispositions and somatic mutations in late-onset AD, while also considering the importance of precision diagnostics, novel therapeutic options, and prevention strategies. The better understanding of the etiology of the more common late-onset form of the disease would benefit from improved GWAS and the introduction of analysis of brain-derived cell-free DNA in plasma. Future GWAS surveys should be performed on understudied ethnic groups as well as with more precisely defined controls matched by sex, age, as well as various environmental and exposure factors. Additionally, these associations should be replicated and validated across diverse populations, followed by functional studies to enhance our understanding of pathophysiology [193]. GWAS data may contribute to the introduction of a polygenic risk score for individualized risk prediction, which can be used to mitigate genetic disease risk by lifestyle and behavioral factors [168]. Pinpointing the heritability of Alzheimer’s disease can contribute to detection of new therapeutic targets that offer fresh opportunities for personalized therapy. The implementation of liquid biopsy by analyzing brain-derived cell-free DNA in plasma for early detection and diagnosis of Alzheimer’s disease requires introduction of standardized protocols and more sensitive techniques to detect cfDNA before it can be reliably used in clinical settings. The analysis of brain-derived cell-free DNA holds a promise for novel blood biomarkers which can be more scalable than CSF and brain imaging markers [168].

The integration of the accomplishments of genetic and other research studies with clinical and institutional experience can lead to significant progress in the deeper understanding of Alzheimer’s diseases and development of more effective therapies.

## Figures and Tables

**Figure 1 ijms-26-07819-f001:**
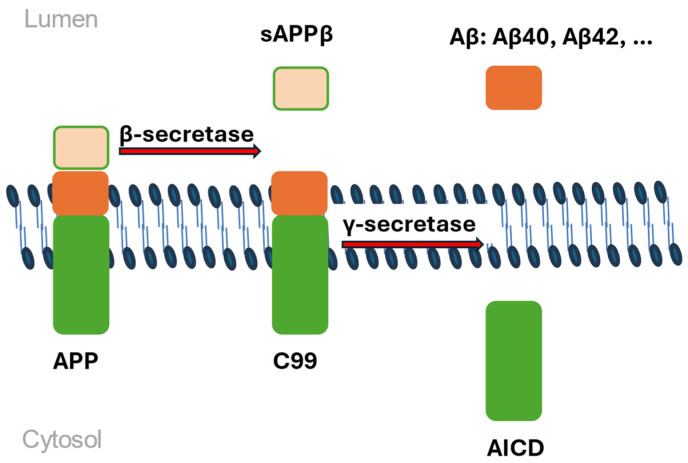
The amyloidogenic pathway. APP is cleaved by β-secretase, generating a soluble extracellular fragment -sAPPβ and a 99-residue, membrane-bound C-terminal fragment—C99. Subsequently, γ-secretase cleaves C99 to release Aβ peptides into the extracellular space and the APP intracellular domain into the cytoplasm (AICD). Mutant PSEN1/2 leads to reduced γ-secretase activity, mutant PSEN2 can disrupt the fusion of the γ-secretase complex.

**Figure 2 ijms-26-07819-f002:**
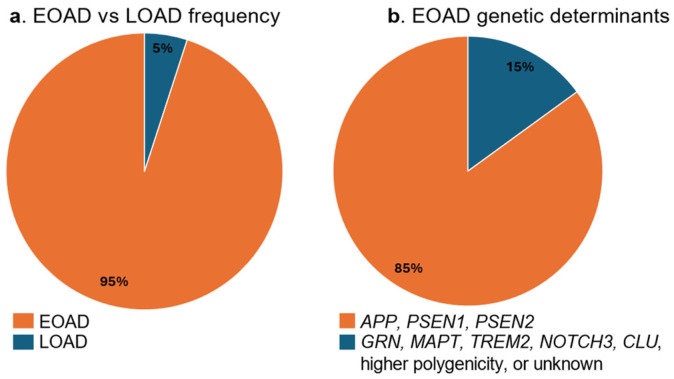
(**a**) Proportion of early-onset and late-onset cases in Alzheimer’s disease; (**b**) 10–15% of mutations causing EOAD are identified in the *APP*, *PSEN1*, and *PSEN2* genes. The remaining cases are due to mutations in various other genes, including *GRN* (progranulin), *MAPT* (microtubule-associated protein tau), *TREM2* (triggering receptor expressed on myeloid cells 2), *NOTCH3* (neurogenic locus NOTCH3 homolog), *CLU* (clusterin), higher polygenicity or the genetic determinant remains unknown.

**Figure 3 ijms-26-07819-f003:**
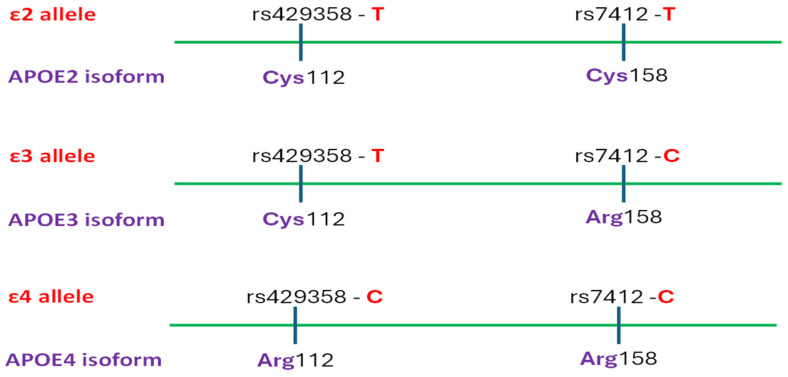
Three common APOE isoforms/alleles and their defining amino acids/polymorphism. Each haplotype is responsible for a different type of the synthesis of a different isoform of the APOE peptide. The ε2 allele is considered a protective allele, ε3 is considered as neutral, whereas ε4 is considered as a risk allele for the development of Alzheimer’s disease.

**Table 1 ijms-26-07819-t001:** Summary table of EOAD- and LOAD-associated genes.

Gene Name	Biological Function	Prevailing Mutation Type	Mechanism Relevant to AD Pathogenesis	Reference
Amyloid precursor protein (*APP*)	Involved in neurite growth, neuronal adhesion	Mainly missense mutations and gross insertions/duplications	Increased total Aβ or Aβ42 levels or Aβ fibrillogenesis	[37]
Presenilin 1 (*PSEN1*)	Catalytic subunit of the gamma-secretase complex	Missense mutations, small insertions, deletions, and genomic deletions, complete penetrance	Cause of the most severe forms of EOAD (as early as 24 years)	[37]
Presenilin 2 (*PSEN2*)	Probable catalytic subunit of the gamma-secretase complex	Missense/nonsense; incomplete penetrance	Found in dementia-associated disorders; EOAD, but later than *PSEN1*	[37]
Triggering receptor expressed on myeloid cells-2 *(TREM2)*	Receptor for Aβ42; mediates multiple pro-inflammatory cytokine processes	Missense/nonsense	Mutated tau accelerates the neurodegenerative process	[54]
Apolipoprotein E (*APOE*)	Binds lipids to form lipoproteins, central role in CNS lipid transport	ε4 allele	Increased intra-neuronal accumulation of Aβ and plaque deposition	[56]
Microtubule associated protein tau (*MAPT*)	Assembly and stabilization of microtubules	Missense	Accelerates the neurodegenerative process in AD	[54]
Bridging integrator 1 (*BIN1*)	Membrane tubulation, endocytosis and intracellular endosome trafficking	Missense/nonsense	Intracellular beta-amyloid accumulation and early endosome enlargement	[62]
Siglec-3 (sialic acid binding Ig-like lectin 3) (*CD33*)	Negative regulation of cytokine production	Missense/nonsense	Decreased Aβ42 uptake and increased expression of full-length *CD33* and *TREM2* in monocytes	[65]
Complement receptor 1 (*CR1*)		Missense/nonsense, gross insertions or duplications	Aβ accumulation in brain	[66]
Phosphatidylinositol-binding clathrin-assembly protein (*PICALM*)	Clathrin-mediated endocytosis, membrane repair of synaptic vesicles	Missense/nonsense, regulatory	Aβ production and clearance, tau-mediated neurodegeneration	[70]
ATP-Binding Cassette Transporter 7 (*ABCA7*)	Lipid transport and immune responses	Missense/nonsense	Impaired clearance of amyloid-beta	[74]
Membrane-spanning 4-Domains A4A, A4E, and A6E, respectively (*MS4A4A*, *MS4A4E*, and *MS4A6E*)	Proteins with four or more transmembrane domains	Missense/nonsense	Disrupted clearance process, amyloid plaque buildup, and increased neuroinflammation	[78]
Sortilin-related Receptor 1 (*sorLA*)	Type 1 transmembrane protein involved in regulating APP intracellular trafficking and processing	Missense/nonsense	Truncating mutations are shown to be highly pathogenic	[80]
Zinc finger CW-type PWWP domain protein 1 *(ZCWPW1)*	Modulates epigenetic regulation	Point mutations	Protective and risk effect depending on population background	[85]
Disintegrin and metalloproteinase domain-containing protein 10 (*ADAM10*)	*α-secretase* that cleaves APP in the non-amyloidogenic pathway	Missense/nonsense	Can lead to age-related downregulation of α-secretase	[74]

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
