# Peer review of "Recent Advances and Future Directions in Alzheimer’s Disease Genetic Research"

_ijms, 2025, doi:10.3390/ijms26167819_

Round 1

Reviewer 1 Report

Comments and Suggestions for Authors

Overall this comprehensive review is well thought but I would recommend some suggestions to improve the interest and strength of state of art of this important research theme.

First and foresmost, I suggest some changes in the main skeletal division:

1. Introduction (I recommend elaborating the goal of this research in an assertive and clear sentence at the end of the introduction to contextualize and direct the reader towards the main and differential point from other reviews.)
2. Ethiological hypothesis of Alzheimer's Disease
 2.1 Cholinergic Hypothesis
 2.2 Amyloid hypothesis (I suggest developing a sentence about the Astrocyte inflammation and hypertrophy near formed plaques. Suggested literature to help:  https://doi.org/10.3390/cells9112415;  https://doi.org/10.1002/glia.24677; DOI: 10.1016/j.molmed.2018.11.006; https://doi.org/10.1007/s00401-021-02372-6)
 2.3 I suggest including here instead of point 5: Genetic ethiology
  2.3.1 Genetics of early-onset Alzheimer’s disease
  2.3.2. Genetics of late-onset Alzheimer’s disease
  2.3.3. Somatic mutations in Alzheimer’s disease 
 2.4. Propose a new subitem: Microbiome (https://doi.org/10.1016/j.neures.2022.05.003; https://doi.org/10.3390/jcm13144130; https://doi.org/10.3390/jcm10112358; https://doi.org/10.3390/microorganisms8040493;  https://doi.org/10.3390/microorganisms8020199; DOI:https://doi.org/10.1007/s11011-023-01248-w; https://doi.org/10.3390/foods14091559; https://doi.org/10.3390/healthcare13010074;https://doi.org/10.1016/j.jnutbio.2024.109813; https://doi.org/10.1007/s42452-025-06821-9; https://doi.org/10.3390/microorganisms13040814) and Metabolomics (https://doi.org/10.3389/fnagi.2025.1530046; DOI:https://doi.org/10.1186/s12967-025-06148-4; https://doi.org/10.1016/j.arr.2025.102739)
3. Stages and symptoms (leave a suggestion to complete this segment: https://doi.org/10.48550/arXiv.2501.05852)
4 Diagnostic (I proposed to subdivide the current text in three parts)
 4.1.Neurobehavioural and socio-cognitive assessment 
 4.2.Laboratory markers and Imaging Tools
 4.3 Current and Future Digital approaches
 4.4 Molecular diagnostic Tools (such as Circular RNA (circRNA): DOI:https://doi.org/10.1007/s12035-024-03977-0; https://doi.org/10.1039/D4NR04394D)
5. AD Treatment
 5.1 Contemporary and Classical Treatment ( maintain Cholinesterase inhibitors + Anti-NMDA + add innovative agents like Regenerate mitochondria (https://doi.org/10.54254/2753-8818/33/20240896),  maintainImmune System-Modulating Drugs, add nanobodies etc etc (https://doi.org/10.3390/ph17060741). It would be also interesting to add how new molecular approaches are inciding and complementing to plant therapy (https://doi.org/10.3389/fmed.2023.1335512); I also suggest including a sentence dedicated to holistic approaches such as diet and lifestyle)
 5.2. Future genetic approaches (Include gene therapy and  Gene-editing technology; If the authors agree, I also suggest including mRNA Vaccine for Alzheimer's Disease: doi: 10.3390/vaccines12060659, doi: 10.3390/pharmaceutics13091397; https://doi.org/10.1039/D4NR04394D)
6. Conclusions

Second: Please verify if references citation is in accord to IJMS requirements and make it uniform throughout the manuscript (https://www.mdpi.com/journal/ijms/instructions). Note: In the text, reference numbers should be placed in square brackets [ ], and placed before the punctuation; for example [1], [1–3] or [1,3]. For embedded citations in the text with pagination, use both parentheses and brackets to indicate the reference number and page numbers; for example [5] (p. 10). or [6] (pp. 101–105).

The manuscript cites many old references I tried to research to help finding some references to add a more updated support to this exhaustive work. I do understand and respect that the authors tried to portray a faithful evolution on the history of identification, diagnostic and treatment of Alzheimer's so I just recommend adding new referencse instead of replacing all the old references.

Abstract: I kindly suggest removing the spaces between paragraphs. 
If possible it would be interestint to (1) clear indicate the goal of this review.
Moreover, I kindly suggest including a graphical abstract. It would catch the attention of readers to help to disseminate your scientific production.

Overal manuscript corrections and/or references suggestions (the authors are free to choose the most convenient recent references for their work):

Please, make sure gene names are italicized through all the text.

Lines 14- 15: Please correct: "which account for 55% of cases; to multiple (...)" to "which account for 55% of the cases, to multiple (...)"
Line 16: Please correct "in 10-15% of cases" to "in 10-15% of the cases"
Lines 33-34: I kindly suggest paraphrasing "and is often accompanied and usually preceded by behavioural and psychological symptoms (...)" to " and it is usually preceded and often accompanied by behavioural and psychological symptoms (...)"
Line 35: Would it be possible to add another more recent reference besides this one from 2012?
Line 45: Please provide the respective references for this information.
Line 72: Would it be possible to add more recent references. I leave only some suggestions to help (the authors are free to choose the most convenient for their work): DOI: 10.1016/j.celrep.2025.115249; https://onlinelibrary.wiley.com; doi/pdf/10.1111/jnc.70054.
Line 75: Please verify more recent references to support this information. Some suggestions: https://doi.org/10.1007/s00401-015-1392-5; DOI: https://doi.org/10.2174/0115672050306008240321034006.
Line 81: Some recent reference suggestions: https://doi.org/10.3390/jcm13051338;  https://doi.org/10.1038/s41392-024-01911-3.
Line 83: Possible adding reference: DOI: 10.1016/j.biopsych.2024.03.004; https://doi.org/10.1016/B978-0-443-19088-9.00009-3.
Line 90: Please add some reference such as  https://doi.org/10.1111/jnc.16183 to information mentioned in "located on chromosome 21."
Line 93: I kindly suggest adding some recent references to (Nistor et al., 2007), like DOI:https://doi.org/10.1007/s00401-025-02844-z; https://doi.org/10.1002/alz.70158
Line 111: I kindly recommend citing a recent reference at the end of the sentence : "(...)of a single tau gene located on chromosome 17."
Suggestion:  https://doi.org/10.1002/alz.13784.
Line 112: I kindly suggest citing a recent reference for this information at the end of the sentence " (...)while the remaining three have four.. My suggestion: https://doi.org/10.1371/journal.ppat.1012926
Line 120: Suggestion to add to (Vilchez et al., 2014) as a more recent reference: https://doi.org/10.1007/s10916-023-01906-7.
Lines 157-158: I recommend adding a recent reference to the end of the sentence: "serotonin-reuptake inhibitor (e.g. escitalopram) can be applied."             Suggestion: https://doi.org/10.3390/ijms25105169.
Line 160: I recommend adding also a recent reference to "(...)risks of complications and death (Aupperle, 2006)." Suggestion: DOI: 10.1016/j.ypsc.2025.01.002.
Line 180: I also kindly recommend adding more assessment tools, depending on different societies/countries: Hasegawa Dementia Scale-Revised, Clock Drawing Test and Cube Copying Test. Also consider adding quality of life assessments. 
Suggestion for references:  https://doi.org/10.1111/ggi.14678;  https://doi.org/10.1002/alz.087484;  (Also, JAQUA, Ecler Ercole; TRAN, Mai-Linh N.; HANNA, Mary. Alzheimer disease: Treatment of cognitive and functional symptoms. American Family Physician, 2024, 110.3: 281-293.)
Line 181: I would kindly suggest mentioning, along the commonly used tools for quantifying these deficits,the updated tries and studies of implemmenting digital cognitive assessment for Alzheimer disease and related dementias in primary care. Possible references: DOI: https://doi.org/10.1370/afm.240293; https://doi.org/10.1007/978-3-031-53976-3_16;  https://doi.org/10.1002/alz.087092; https://doi.org/10.1016/j.arr.2024.102497.
Line 191 and 195: I suggest adding the reference  https://doi.org/10.1002/kjm2.12870 to complement both citations to the existing references (line 191 and line 195)
Line 212: Also recommend citing a current effort for early diagnosis such as mentioned in https://doi.org/10.1371/journal.pone.0310966.
Line 222: I kindly suggest complemmenting references with more recent articles such as:https://doi.org/10.1002/alz.14486 and/or DOI: https://doi.org/10.1007/s11357-024-01113-3.
Lines 223-495: Overall it is well explained and fundamented but it would be interesting to further explain molecular mechanisms involved by the genes indicated in the text. Some references to help the authors: https://doi.org/10.1371/journal.pgen.101163;  https://doi.org/10.1002/alz.14486; https://doi.org/10.3390/diseases12060110; https://doi.org/10.3390/bioengineering11010045; DOI: 10.4103/1673-5374.382232; DOI:https://doi.org/10.1038/s41593-024-01669-5.
To simplify the mechanisms explanations, the authors could develop a synthetic table of mechanisms involved in early onset versus late onset,to be added as supplementary material.
Figure 2b: Please enlarge the circle to the same size of the 2a.
Line 548: Please add the respective reference.
Adapt conclusion to new information added in the text, according to the suggestions.

Thank you for your attention. Congratulations for the excellent research work developed so far.

Author Response

Comments 1:

Introduction (I recommend elaborating the goal of this research in an assertive and clear sentence at the end of the introduction to contextualize and direct the reader towards the main and differential point from other reviews.)

Response 1: Thank you for pointing out this missing piece in the Introduction. We agree with the comment, therefore, we have added a paragraph at the end of the Introduction section including proposals from Reviewer 2, clarifying the goal of the review. The paragraph is as follows:

“The increasing prevalence and socio-economic burden of Alzheimer’s disease require novel and integrated approaches for prevention, early diagnosis and treatment. In this context, this review seeks to provide a comprehensive overview of the genetic architecture of Alzheimer’s disease, with a primary emphasis on the underlying molecular mechanisms underlying both the familial and the sporadic forms. It synthesizes recent key discoveries that may inform the development of future diagnostic strategies and therapeutic interventions. Given the ongoing advances in the field, this topic is particularly timely and is intended to serve as a valuable resource for clinicians and researchers engaged in Alzheimer’s disease research and care.”

Comments 2: 2. Ethiological hypothesis of Alzheimer's Disease
 2.1 Cholinergic Hypothesis
 2.2 Amyloid Hypothesis

Response 2: Agree. We have renumbered the subsections of this section accordingly.

Comments 3: I suggest developing a sentence about the Astrocyte inflammation and hypertrophy near formed plaques. Suggested literature to help: https://doi.org/10.3390/cells9112415; https://doi.org/10.1002/glia.24677;DOI:10.1016/j.molmed.2018.11.006;https://doi.org/10.1007/s00401-021-02372-6)

Response 3: Agree. We have included a sentence for the role of astrocytes in neuroinflammation “Various reactive astrocyte sub-types have been found in regions with Aβ or tau pathology, which is probably due to their role of in neuroinflammatory changes in AD defined by the release of cytokines, chemokines, reactive oxygen species and other inflammatory factors.” and we have included astrocytes in cell types with high expression of genes with AD risk variants in the previous sentence, based on the information from the suggested references.

Comments 4: Propose a new subitem Microbiome (https://doi.org/10.1016/j.neures.2022.05.003;https://doi.org/10.3390/jcm13144130; https://doi.org/10.3390/jcm10112358; https://doi.org/10.3390/microorganisms8040493;  https://doi.org/10.3390/microorganisms8020199;DOI:https://doi.org/10.1007/s11011-023-01248-w;https://doi.org/10.3390/foods14091559;https://doi.org/10.3390/healthcare13010074;https://doi.org/10.1016/j.jnutbio.2024.109813;https://doi.org/10.1007/s42452-025-06821-9;https://doi.org/10.3390/microorganisms13040814) and Metabolomics (https://doi.org/10.3389/fnagi.2025.1530046;DOI:https://doi.org/10.1186/s12967-025-06148-4;https://doi.org/10.1016/j.arr.2025.102739)

Response 4: Agree. We have added the two proposed subsections - Microbiome and Metabolomics, as 2.4 Microbiome and 2.5 Metabolomics. This subsequently influenced a change in the numbering of sections which were changed accordingly. We have used the proposed literature. The subsections are, as follows:

2.4. Microbiome

The gut-brain axis is a bidirectional communication network that integrates neural and humoral signals, playing a key role in both neurological and gastrointestinal function. Emerging research highlights the gut microbiota as a genetically influenced system that contributes to host physiology through metabolites such as short-chain fatty acids, peptides, and neurotransmitters. Genetic variation affecting microbiome composition may influence susceptibility to neurodegenerative and psychiatric disorders, including Alzheimer’s disease and depression. Understanding the genetic regulation of gut-brain interactions offers a promising avenue for elucidating disease mechanisms and identifying novel therapeutic targets. Growing evidence supports a strong association between the gut microbiota (GM) and cognitive as well as mental health impairments. Experimental models, such as germ-free mice and antibiotic-induced dysbiosis, have demonstrated disruptions in the production of key neuroactive metabolites—including brain-derived neurotrophic factor (BDNF), gamma-aminobutyric acid (GABA), N-methyl-D-aspartate (NMDA) receptors, and tryptophan—that are essential for cognitive function. Studies of neurological and psychiatric disorders marked by cognitive deficits, such as Parkinson’s disease, Alzheimer’s disease, schizophrenia, and major depressive disorder, suggest that alterations in GM composition may play a role in their pathogenesis. Consequently, modulation of the gut microbiota represents a promising strategy for enhancing cognitive function in both clinical and experimental contexts. Emerging research identifies the oral microbiome as a key modulator of cognitive health and Alzheimer’s disease (AD) risk. Oral dysbiosis promotes systemic inflammation, disrupts the blood–brain barrier, and contributes to neuroinflammation—processes central to AD pathogenesis. The oral–brain and oral–gut–brain axes mediate immune, neural, and endocrine signaling across the mouth, gut, and brain. Pathogens such as Porphyromonas gingivalis, along with virulence factors like LPS and gingipains, and microbial metabolites (e.g., SCFAs, peptidoglycans), can amplify inflammatory responses and affect brain function.Host genetics shape and respond to these microbial shifts. For instance, APOE ε4 carriers show altered oral and gut microbiota profiles, including reduced microbial diversity and increased abundance of pro-inflammatory species. Additionally, polymorphisms in TLR4, a key innate immune receptor, influence individual responses to LPS from oral pathogens, modulating the intensity of neuroinflammatory cascades. Environmental exposures—diet, pH, medications, smoking, alcohol, and oral hygiene—further interact with host genotype to influence microbial composition and cognitive outcomes. Understanding gene–microbiome interactions is essential for identifying at-risk individuals and developing targeted interventions. Strategies such as probiotics, prebiotics, and dietary modulation may mitigate genetic risk and delay cognitive decline. Longitudinal studies are needed to clarify causal pathways linking host genetics, the oral microbiome, and neurodegeneration. For millennia, humans have maintained a symbiotic relationship with diverse bacterial communities known as the microbiota, which play a crucial role in maintaining homeostasis, health, and various physiological functions. This paper examines the relationship between gut microbiota (GM) and neurodegenerative processes, with a focus on how microbial imbalances (dysbiosis) impact the human nervous system. Under normal conditions, there is molecular communication between microbial populations and neural cells; disruption of this interaction can contribute to the onset of pathological conditions. We review key molecular mechanisms altered during dysbiosis and the associated neurological disorders. Understanding these mechanisms may reveal novel therapeutic targets to mitigate neuroinflammation and neurodegeneration. The human intestine hosts one of the most densely populated microbial ecosystems in nature. It is estimated that approximately 100 trillion microorganisms reside in the adult gut—outnumbering the total number of human cells, which is around 10^{13}. Over the past decade, research into the link between neurodegenerative diseases and the microbiome—especially the gut and oral microbiomes—has grown substantially. Amyloid beta (Aβ) plaque formation, hyperphosphorylated tau proteins, and neurofibrillary tangles (NFTs) in brain tissue are classic pathological features of Alzheimer’s disease (AD). The amyloid cascade hypothesis (ACH), which has dominated AD research for two decades, proposes that Aβ accumulation is the primary trigger of neurodegeneration. However, recent findings challenge this view by showing that neuronal injury biomarkers can occur independently of Aβ, suggesting the need to reconsider the disease cascade and explore alternative therapies.These inflammatory lesions cause progressive neuronal loss in vulnerable brain regions, leading to cognitive decline in AD. Neuroinflammation is linked to the clearance of Aβ aggregates by microglial phagocytosis. While the classical ACH focuses on Aβ-42 as the key driver of AD pathology, it fails to account for the disorder’s complexity. For example, Aβ and tau may be byproducts rather than causes of neurodegeneration; the direct causal link between Aβ and tau is uncertain; amyloid plaques and NFTs may not directly cause dementia; APP-based transgenic models do not fully replicate AD pathology; and therapies targeting the ACH have largely been unsuccessful. These issues have prompted modifications to the ACH for a more comprehensive understanding of AD pathogenesis.Neuropathologically, AD is marked by neuronal loss, extracellular β-amyloid plaques, and intracellular neurofibrillary tangles composed of hyperphosphorylated tau. Aβ is produced through proteolytic cleavage of the β-amyloid precursor protein (APP). In individuals with Down syndrome (DS), who carry an extra copy of APP, increased APP expression leads to elevated Aβ deposition and typical AD neuropathology. Familial early-onset AD has been linked to missense mutations in APP, while the APP A673T variant has been identified as protective, reducing AD risk. Cellular and animal studies show that risk-associated APP mutations increase total Aβ, Aβ42 levels, or fibrillogenesis, whereas protective variants reduce these factors. Together, these findings support the Aβ hypothesis and suggest that reducing Aβ levels or aggregation may help prevent or lower AD risk.AD’s multifactorial nature complicates pinpointing its exact cause, but risk factors include genetic factors, such as APP mutations leading to familial and sporadic AD, and non-genetic factors like aging, immune dysfunction, anatomical degradation, and environmental influences. While the gut microbiome’s role in neurodegeneration is well studied, the oral microbiome—directly exposed to pathogens and systemic signals—has received less attention, despite evidence linking bacteria like Porphyromonas gingivalis to increased AD risk. The oral–brain and oral–gut–brain axes mediate communication through neural, immune, and endocrine pathways, where dysbiosis can trigger systemic inflammation and contribute to neurodegeneration and related systemic diseases that exacerbate cognitive decline. An integrated healthcare approach targeting both oral and gut microbiome health is crucial for maintaining neurological function, as this review highlights the impact of oral microbial balance and its interaction with the gut microbiome on cognitive outcomes. Neurodegeneration can be driven by microbiota through multiple pathways, with gut dysbiosis promoting local and systemic inflammation that disrupts the intestinal epithelial barrier and increases brain permeability, leading to neuroinflammation and neuronal dysfunction. Elevated levels of bacterial endotoxins like lipopolysaccharide (LPS) have been found in Alzheimer’s disease (AD) brains, where LPS co-localizes with β-amyloid (Aβ) plaques, enhances Aβ fibrillation, and activates inflammatory signaling pathways such as NF-κB, contributing to nerve inflammation. Chronic systemic exposure to LPS in animal models induces cognitive impairments, β-amyloid accumulation, and microglial activation, further exacerbating neurodegeneration. Additionally, bacterial amyloid peptides from families like Akkermansiaceae and Enterobacteriaceae may promote Aβ aggregation, while beneficial microbes such as Bifidobacteriaceae are reduced in AD, highlighting the complex role of microbiome balance in disease progression.

2.5 Metabolomics

Gut microbiota dysbiosis—characterized by a reduced abundance of beneficial bacterial species—has been consistently observed in patients with Alzheimer’s disease. Given the growing recognition of the microbiome’s role in modulating neurodegenerative processes, recent studies have explored microbial and plant-derived compounds as potential therapeutic agents. This interest is further supported by metabolomic analyses, which reveal that alterations in gut microbial composition can influence the production of neuroactive metabolites, such as short-chain fatty acids and tryptophan derivatives, that impact brain health. These findings underscore the value of integrating microbiome research with metabolomics to identify novel biomarkers and therapeutic targets for AD. Alzheimer’s disease (AD) is marked by the buildup of amyloid beta plaques and neurofibrillary tangles of hyperphosphorylated tau protein. One study used computational methods to assess natural neem compounds (limonoids) and gut microbiome metabolites for their ability to inhibit key AD targets. Molecular docking of around 200 neem phytochemicals and 9 microbial metabolites was performed against beta-secretase 1 (BACE1), gingipain cysteine protease, and tau oligomerization receptors, with blood-brain barrier (BBB) permeability predicted using six molecular descriptors. Results showed that limonoids—particularly rutin, 7-benzoylnimbocinol, and tirucallol—exhibited strong binding affinities to these targets, surpassing melatonin by over 30%, though their BBB penetration may require advanced delivery methods. Among microbiome metabolites, melatonin showed moderate binding across all targets. These findings suggest limonoids as promising multi-target AD inhibitors and support further development of nanocarrier-based strategies to enhance their brain delivery. Nervonic acid, found in breast milk, fish oil, and certain vegetable oils, is vital for human nervous system development. In this study, the Morris water maze test and pathological analysis demonstrated that nervonic acid improved cognitive deficits and brain nerve damage in AD rats. Sequencing revealed that nervonic acid increased beneficial bacteria like Lactobacillus and Bacteroides while reducing Pseudomonadaceae_Pseudomonas. Additionally, it modulated short-chain fatty acid production, altered 29 fecal metabolites, influenced the metabolism of linoleic acid, α-linolenic acid, arachidonic acid, and sphingolipids, and regulated metabolic enzyme activity. Nervonic acid (C24:1Δ15; cis-tetracos-15-enoic acid) is a functional very long-chain monounsaturated fatty acid known for its significant biological roles in brain development and overall health. Nervonic acid was first found in human and mammalian brains and was extracted from shark oil. Studies have shown that the development of neurological disorders is closely related to nervonic acid deficiency, such as AD and Parkinson's disease. Nervonic acid is widely researched as a dietary supplement and a key ingredient in certain edible oils. Naturally present in breast milk, it supports infant neurological development and is therefore commonly added to commercial infant formulas as a functional additive. One study found that feeding nervonic acid-containing vegetable oil to rats helped improve the animals’ learning and cognitive abilities. These findings suggest that nervonic acid has potential as a functional food for treating neurological disorders. Our previous research demonstrated that nervonic acid exerts therapeutic effects on Alzheimer’s disease by modulating metabolic pathways, including tryptophan metabolism, which is closely linked to gut microbiota health. In light of this, the research group proceeded to analyze fecal metabolites and gut microbiota, recognizing their pivotal role in neurodegenerative diseases such as Alzheimer’s disease (AD). According to their findings, disturbances in the gut microbiota contribute to AD pathogenesis through mechanisms including neuroinflammation, immune dysfunction, and neuronal cell death. Their results emphasized a strong link between gut microbial imbalance and brain disorders. Notably, they reported that such microbial disturbances can occur prior to the clinical onset of AD, suggesting their potential as early biomarkers. The gut microbiota influences brain function in part through the production of small molecule metabolites, notably short-chain fatty acids (SCFAs). SCFAs—such as acetate, propionate, and butyrate—are key microbial byproducts that play essential roles in gut-brain communication. Research has demonstrated that both the levels of SCFAs in the gastrointestinal tract and the abundance of SCFA-producing bacteria are strongly linked to the onset and progression of neurological disorders. These findings highlight the importance of microbial metabolites in modulating brain health and disease.

Comments 5: Stages and symptoms (leave a suggestion to complete this segment: https://doi.org/10.48550/arXiv.2501.05852)

Response 5: Agree. We have, accordingly, completed the segment by adding the following sentence from the recommended reference: “The efficient and accurate early diagnosis and staging of Alzheimer’s disease will benefit greatly from further implementation of the currently developing machine-learning frameworks, which can be utilized region-specific structural MRI patterns as well as other diagnostic tool data.

Comments 6: 4 Diagnostic (I proposed to subdivide the current text in three parts)
4.1.Neurobehavioural and socio-cognitive assessment
4.2.Laboratory markers and Imaging Tools

4.3 Current and Future Digital approaches
4.4 Molecular diagnostic Tools (such as Circular RNA (circRNA): DOI:https://doi.org/10.1007/s12035-024-03977-0; https://doi.org/10.1039/D4NR04394D)

Response 6: Agree. We have divided part 4. Diagnostics into the 4 suggested subsections and we used the originally proposed text as information for the introduction to the paragraph, as well as information for  4.1.Neurobehavioural and socio-cognitive assessment. We then used the suggested literature and more to summarise existing information for the three other subsections. The changes are, as follows:

4. Diagnostics Diagnosing Alzheimer’s disease involves a thorough, multidisciplinary approach that encompasses clinical evaluation, cognitive testing, biomarker investigation, and neuroimaging techniques. The diagnostic process typically begins with a detailed exploration of the patient’s medical and family history, with attention to genetic risk factors—most notably the apolipoprotein E (APOE) ε4 allele—and any coexisting medical conditions that could contribute to cognitive impairment (Bomasang-Layno & Bronsther, 2021). This initial assessment is typically followed by a physical and neurological examination aimed at detecting motor, sensory, or other neurologic signs that may aid in distinguishing Alzheimer’s disease from other potential diagnoses.

4.1.Neurobehavioural and socio-cognitive assessment

Cognitive and neuropsychological assessments play a pivotal role in the diagnosis of Alzheimer’s disease, as they help detect impairments across multiple cognitive domains such as memory, executive functioning, attention, language, and visuospatial abilities. Commonly used tools for quantifying these deficits include the Mini-Mental State Examination (MMSE) and the Montreal Cognitive Assessment (MoCA) (Nasreddine et al., 2005). There are also multiple country-specific assessment tools, like the Linus Health Core Cognitive Evaluation and the Digital Clock and Recall DCAs by Linus Health, Inc; as well as options for questionnaires in the native language of specific patients. For instance, a Japanese study comments on the common use of the Wisconsin Card Sorting Test and the Trail Making Test, as well as the Clock Drawing Test (CDT) and the Cube Copying Test (CCT) for screening. Multiple other questionnaires and scales are also being employed in the screening process for Alzheimer’s disease according to the same study with the Neuropsychiatric Inventory, the Self-Rating Depression Scale and the Geriatric Depression Scale being a few.

Additionally, quality of life assessments can also be employed for the neurobehavioural and cognitive assessment of Alzheimer’s disease. According to the Alzheimer’s Society of the United Kingdom, have pinpointed several exemplary questionnaires or instruments that can be used to measure the quality of life of people living with the condition.  The three questionnaires are: DEMQOL, QoL-AD and DQoL.

In summary, neuropsychiatric evaluation is essential due to the frequent occurrence of behavioural and psychological symptoms in individuals with AD, such as depression, apathy, anxiety, and agitation. Artificial intelligence can also have an enormous impact on AI diagnosis and management.

4.2.Laboratory markers and imaging tools

Different laboratory tests, such as blood analyses and cerebrospinal fluid (CSF) examination, are utilized to rule out alternative causes of cognitive impairment and to provide additional diagnostic support for Alzheimer’s disease. Key CSF biomarkers—namely, reduced concentrations of amyloid-beta 42 (Aβ42) alongside increased levels of total tau and phosphorylated tau—have shown strong sensitivity and specificity in identifying Alzheimer’s pathology, making them valuable tools in both clinical practice and research contexts (Jack et al., 2018).

Neuroimaging serves as a valuable adjunct in the diagnostic evaluation of Alzheimer’s disease. Structural imaging techniques, such as magnetic resonance imaging (MRI) and computed tomography (CT), facilitate the detection of brain atrophy patterns that are indicative of AD, especially within the medial temporal lobe (Frisoni et al., 2010). Functional imaging techniques, including fluorodeoxyglucose positron emission tomography (FDG-PET), can identify regions of reduced metabolic activity, most notably in the temporoparietal cortex (Foster et al., 2019). Beyond structural imaging, molecular techniques like amyloid and tau PET scans can be utilized to detect the fundamental pathological markers of Alzheimer’s disease in vivo. Utilizing a combination of these methods improves diagnostic precision, especially in the early, preclinical, and prodromal stages of the disease.

 4.3 Current and future digital approaches

The integration of digital technologies with genomic research has opened new avenues for the early diagnosis and risk stratification of Alzheimer’s disease (AD). In the context of human genetics, several digital approaches have already been deployed, while others are under development and show promise for future clinical translation. Current digital approaches include polygenic risk scores (PRS), next-generation sequencing (NGS) with AI-based interpretation, digital biobanks and integrated genomic platforms as well as many more. PRS are algorithmically derived estimates of genetic liability to AD, calculated by aggregating the effects of numerous single nucleotide polymorphisms (SNPs) identified through genome-wide association studies (GWAS). These scores provide a quantitative measure of an individual’s inherited risk and have shown utility in stratifying at-risk populations, including individuals who do not carry the APOE ε4 allele. The application of NGS technologies, such as whole-exome and whole-genome sequencing, has enabled the identification of both common and rare genetic variants associated with AD. Machine learning algorithms now assist in the interpretation of large-scale sequencing data, allowing for the automated classification of variants based on predicted pathogenicity and disease relevance. Large-scale digital biobanks (e.g., UK Biobank, Alzheimer’s Disease Neuroimaging Initiative [ADNI]) combine genetic, clinical, and phenotypic data from thousands of individuals. These platforms employ bioinformatic pipelines to facilitate the discovery of novel AD-related genetic variants and provide a resource for cross-validation of polygenic models. Future digital approaches include digital twin modeling for predictive genomics, CRISPR screening platforms with AI integration and more. A digital twin refers to a dynamic computational replica of an individual, constructed using multi-omic, clinical, and environmental data. In the context of AD, digital twin models may offer a personalized framework for simulating disease progression, forecasting therapeutic responses, and optimizing intervention timing based on an individual’s genetic profile. Emerging platforms are beginning to combine CRISPR-Cas9 gene-editing technologies with deep learning models to simulate and predict the phenotypic effects of specific genetic perturbations. These tools could enable high-throughput screening of AD risk genes, facilitating the identification of novel targets and pathways implicated in disease pathogenesis.
 4.4 Molecular diagnostic tools

Messenger RNA (mRNA) therapy is a cutting-edge biomedical strategy that delivers genetic instructions for the in situ production of therapeutic proteins. By enabling ribosomes to synthesize specific proteins directly within cells, mRNA therapy can restore or introduce functional or immunogenic proteins to address various diseases. Unlike traditional gene therapies that require nuclear entry and pose risks of genomic integration, mRNA operates in the cytoplasm, offering a safer profile. Additionally, protein expression can be finely controlled by modulating the administered mRNA dose and its stability. As a next-generation gene therapy platform, mRNA therapy combines precision, safety, and adaptability, and has already demonstrated its clinical value in the development of COVID-19 vaccines. While acute neurological conditions like stroke often necessitate immediate invasive interventions, chronic neurodegenerative diseases (NDs) and brain tumors progress more gradually and are complicated by factors such as the blood–brain barrier and multifactorial pathology—making them promising candidates for mRNA-based therapeutics. Ongoing research suggests that mRNA therapies could become a transformative tool in both the prevention and treatment of these conditions. This review outlines the current methods for preparing and delivering mRNA drugs, surveys recent advances in mRNA gene therapy for NDs and brain tumors, and discusses prevailing challenges—offering a foundation for future exploration in this emerging field. Early diagnosis and timely intervention are critical for effectively managing Alzheimer’s disease (AD). However, current diagnostic tools and treatment options remain limited, underscoring the urgent need for novel biomarkers and therapeutic targets that can detect and address AD in its earliest stages. Circular RNAs (circRNAs), a unique class of non-coding RNAs characterized by their covalently closed-loop structures, have emerged as promising candidates with potential roles in both the prevention and treatment of various diseases. Recent advances in circRNA research related to AD have yielded several compelling findings—some of which remain underexplored but carry significant scientific and clinical value. This review highlights the current body of research on circRNAs in the context of AD and discusses the potential of circRNA-based approaches for future clinical translation.

Early and accurate diagnosis of Alzheimer's disease is essential for better treatment results and machine learning models are being designed to refine diagnostic accuracy by analyzing diverse data types (Muhammed Niyas & Thiyagarajan, 2023). Delaying the onset of Alzheimer’s disease by just 5 years can have a tremendous effect on the economic burden and is estimated to be able to reduce it by 36% by 2050 (Livingston et al., 2017). Prompt detection allows patients and their families to undertake proactive care planning, gain access to both pharmacological and supportive interventions, and potentially participate in clinical trials targeting disease modification. Furthermore, early diagnosis supports more effective management of comorbid conditions and promotes more efficient use of healthcare resources (Cummings et al., 2019).

Reviewer 2 Report

Comments and Suggestions for Authors Alzheimer’s disease (AD) is the most common form of dementia and a progressive neurodegenerative disorder that currently affects over 50 million individuals worldwide. Clinically, it is characterized by cognitive decline, memory impairment, and behavioral disturbances. Pathologically, AD is defined by the presence of extracellular amyloid-beta (Aβ) plaques and intracellular neurofibrillary tangles composed of hyperphosphorylated tau protein. As global life expectancy continues to rise, the incidence of AD is expected to increase markedly, presenting major challenges for healthcare systems and societal infrastructure. Although age is the most significant risk factor for Alzheimer’s disease, genetic predisposition also plays a vital role in both its early- and late-onset forms. Over the past thirty years, researchers have identified several key genes—such as APP, PSEN1, and PSEN2—that are directly responsible for familial early-onset AD. In contrast, genome-wide association studies (GWAS) have revealed numerous genetic risk loci associated with late-onset Alzheimer’s disease (LOAD), with the APOE ε4 allele being the most prominent. Other implicated genes are involved in critical biological processes, including lipid metabolism, immune system function, and endocytosis. This review examines the genetic landscape of Alzheimer’s disease, with a focus on the molecular mechanisms driving both inherited and sporadic forms. It also highlights recent discoveries that could guide the development of future diagnostic tools and therapeutic approaches. The subject matter is both timely and well-presented, and with minor revisions, the paper has the potential to become a valuable reference for clinicians and researchers working in the field of Alzheimer’s disease genetics.   Comments: While the paper appropriately emphasizes APOE and GWAS findings, it would benefit from a more in-depth discussion of other emerging areas, such as epigenetic regulation, sex-specific differences, and genetic studies in non-European populations, which remain underrepresented yet are crucial for a comprehensive understanding of Alzheimer’s disease. Some terms, such as amyloidogenic and polygenic burden, are used without explanation; consider adding a glossary or providing brief definitions in parentheses. Given the genetic complexity discussed, incorporating several tables would enhance the manuscript’s clarity and impact: A summary table of EOAD and LOAD-associated genes (e.g., APP, PSEN1, PSEN2, APOE, BIN1, MAPT), including columns for gene name, biological function, mutation type, and relevance to Alzheimer’s disease. A table listing key GWAS loci, detailing associated SNPs, odds ratios, and the biological pathways implicated. A table summarizing ongoing or recent clinical trials targeting major pathological pathways such as amyloid, tau, and inflammation. Figure 1. Clearly label the β-secretase and γ-secretase cleavage sites on the amyloid precursor protein (APP). Add descriptive annotations or callouts to indicate how PSEN1 and PSEN2 mutations impact γ-secretase activity and downstream Aβ production.amyloid precursor protein (APP). Olygomer to Oligomer Figure 2. In panel (a), consider adding numerical values or percentages directly on the pie chart to enhance interpretability. In panel (b), the pie chart appears distorted—please adjust the aspect ratio to ensure accurate visual representation. Figure 3. Add a brief explanation of how the depicted polymorphisms influence Alzheimer's disease (AD) risk, either directly within the figure legend or as accompanying figure notes. This will help contextualize the genetic data for readers and clarify the relevance of each variant.

Author Response

Comments 1: While the paper appropriately emphasizes APOE and GWAS findings, it would benefit from a more in-depth discussion of other emerging areas, such as epigenetic regulation, sex-specific differences, and genetic studies in non-European populations, which remain underrepresented yet are crucial for a comprehensive understanding of Alzheimer’s disease.

Response 1: Agree. We have made the suggested additions using appropriate recent literature. The additions are, as follows: The critical role of gene–environment interactions in the multifactorial etiology of Alzheimer’s disease can be mediated by epigenetic mechanisms. The timing of these environmental risk factors for AD appears to be pivotal, with early-life and developmental periods representing windows of heightened epigenetic sensitivity that may shape lifelong AD risk. Genome-wide and gene-specific studies have revealed differential DNA methylation patterns in AD brains and peripheral tissues—such as hypomethylation at APOE and TREM2 loci, hypermethylation of BDNF and SPINT1 promoters, and global methylation shifts linked to APOE ε4 status—that correlate with disease progression and cognitive decline, suggesting their utility as diagnostic or prognostic biomarkers. At chromatin level, AD-related histone acetylation and deacetylation imbalances—mediated by enzymes like histone deacetylases 2 and 6 and sirtuins—have been shown to impair synaptic plasticity, memory formation, and microglial function in both postmortem tissue and animal models, with histone deacetylase inhibitors demonstrating potential cognitive benefits. Furthermore, dysregulation of multiple ncRNA classes—including miRNAs (e.g., miR‑29, miR‑206, miR‑200b/c), lncRNAs (e.g., BACE1-AS, 51A, EBF3-AS), circular RNAs, and alterations in snRNA aggregates—has been implicated in APP processing, amyloid accumulation, tau pathology, and neuroinflammation. Collectively, these interconnected epigenetic mechanisms not only deepen our understanding of AD pathophysiology but also illuminate promising avenues for therapeutic intervention, such as epigenetic drugs targeting DNA methylation, histone modifiers, and ncRNA networks, though translational challenges remain before clinical application can be realized.

As in many other polygenic disorders, most of the studies of their heritability are performed on populations of European descent, as recent genetic studies of Alzheimer’s disease (AD) in non-European populations have revealed both shared and population-specific risk factors, highlighting the importance of inclusive genomic research. In Latin American cohorts, particularly in Mexico and Colombia, autosomal-dominant PSEN1 mutations such as A431E and E280A have been identified as major contributors to early-onset familial AD. Notably, protective variants like APOE3-Christchurch and a rare RELN variant have been observed in Colombian individuals carrying the PSEN1 E280A mutation, significantly delaying disease onset and suggesting genetic resilience mechanisms. In South America, the first genome-wide association study conducted in Argentina and Chile, involving admixed populations, confirmed the known APOE ε4 risk and identified novel loci associated with lysosomal and immune function pathways, including TREM2L, IGH, ADAM17, and PLCG2. A key finding about the predictive power of European-derived polygenic risk scores was found to decline with increasing Native American ancestry, emphasizing the need for ancestry-informed risk modeling. Similarly, GWAS in East Asian populations (Chinese, Korean, and Japanese) have uncovered 26 AD-associated loci, with rare variants such as TREM2 p.H157Y and SHARPIN p.G186R/p.R274W showing significant associations unique to these populations.

Sex-specific differences in Alzheimer's disease (AD) have become a critical area of research, revealing distinct patterns in disease prevalence, progression, and pathology between males and females. Studies consistently report that women are disproportionately affected by AD, exhibiting higher incidence and more rapid cognitive decline compared to men. Biological factors, including hormonal differences such as estrogen decline during menopause, are thought to contribute to this disparity by influencing amyloid-beta accumulation and tau pathology.  Moreover, genetic studies highlight sex-specific effects of the APOE ε4 allele, with female carriers showing greater risk and earlier onset than their male counterparts.  Neuroimaging research further supports these findings, demonstrating sex-dependent variations in brain atrophy and connectivity alterations in AD patients.  Together, these studies underscore the importance of incorporating sex as a biological variable in AD research to improve diagnosis, treatment, and prevention strategies tailored to men and women. Performed GWAS studies of Alzheimer’s disease show that genetic variations differ across ethnic groups (Chen et al., 2021), which highlights the importance of conducting further studies across diverse populations to identify ethnic-specific polymorphisms. Most GWAS to date have primarily been focused on Caucasian populations, leading to potential bias by not including other groups and overlooking possible genetic variations that may contribute to disease risk. For instance, African Americans are twice as likely to develop AD (Bellenguez et al., 2022), emphasizing the need for more studies involving these populations. Similarly, even though women are at a significantly higher risk and experience worse clinical and pathological outcomes (Vila-Castelar et al., 2022) gender-specific genetic factors in AD are still understudied (Nazarian et al., 2019). Some GWAS studies may be biased due to imbalances in sample sizes between controls and AD patients, as well as unaccounted for factors such as comorbidities, gender, age range, environmental exposures, and medication use, which can influence the study results.

Comments 2: Some terms, such as amyloidogenic and polygenic burden, are used without explanation; consider adding a glossary or providing brief definitions in parentheses.

Response 2: Agree. We have added the suggested definition for polygenic burden in the text, lines 172-173. The definition for amyloidogenic pathway already existed in the text - lines 206-207.

Comments 3: Given the genetic complexity discussed, incorporating several tables would enhance the manuscript’s clarity and impact: A summary table of EOAD and LOAD-associated genes (e.g., APP, PSEN1, PSEN2, APOE, BIN1, MAPT), including columns for gene name, biological function, mutation type, and relevance to Alzheimer’s disease. A table listing key GWAS loci, detailing associated SNPs, odds ratios, and the biological pathways implicated.

Response 3: Agree. Both tables have been included in the review - a summary table for EOAD&LOAD-associated genes as Table 1 in the review itself and a table listing key GWAS loci as Supplementary Table 1.

Comments 4: A table summarizing ongoing or recent clinical trials targeting major pathological pathways such as amyloid, tau, and inflammation.

Response 4: Agree. We had already mentioned several key clinical trials in the text - lines 995 to 1007, 1130 to 1135, 1161 to 1170, 1222 to 1224, 1239 to 1248. We have additionally made a table with the most recent clinical trials listed on clinicaltrials.gov as Active, not recruiting, and revolving around the three main pathways for the condition using the integrated Search bar.

Comments 5: Figure 1. Clearly label the β-secretase and γ-secretase cleavage sites on the amyloid precursor protein (APP). Add descriptive annotations or callouts to indicate how PSEN1 and PSEN2 mutations impact γ-secretase activity and downstream Aβ production.amyloid precursor protein (APP).

Response 5: Agree. We have made the suggested correction.

Comments 6: Olygomer to Oligomer

Response 6: Assuming the reviewer meant to suggest we change the spelling of the word, we have adapted the text accordingly.

Comments 7: Figure 2. In panel (a), consider adding numerical values or percentages directly on the pie chart to enhance interpretability. In panel (b), the pie chart appears distorted—please adjust the aspect ratio to ensure accurate visual representation.

Response 7: Agree. We have made the suggested correction.

Comments 8: Figure 3. Add a brief explanation of how the depicted polymorphisms influence Alzheimer's disease (AD) risk, either directly within the figure legend or as accompanying figure notes.

Response 8: Agree. We have made the suggested correction.

Round 2

Reviewer 1 Report

Comments and Suggestions for Authors

Congratulations for your great work! It improved considerably. I would just suggest some minor corrections to make the manuscript appear with a 200% clean presentation:

Table 1. Summary table of EOAD and LOAD-associated genes - Please make sure that references are displayed according to journal's rules or cite with Surname of first author et al., year of publication.
Supplementary file: Supplementary Table 1 - I strongly suggest references are displayed by Surname of first author et al., year of publication, in the respective table column.
Lines 487 - 496: Please include reference at the end of the sentence.
Lines 500-501: Please include reference at the end of the sentence.
Lines 536-542: Please include reference at the end of the sentence.
Lines 551-553: "Additionally, patients diagnosed with AD that are considered dangerous to themselves and/or others are usually hospitalized or institutionalized but rarely tested for additional psychological conditions". I kindly suggest referring as " (...)might be considered (...).

Author Response

Thank you for the prompt and thoughtful feedback on our manuscript, it is greatly appreciated.

We agree with the points raised and have made the effort to address them properly.

Agree. The references in both Table 1 and Supplementary Table 1 have been redacted to align with the IJMS referencing guidelines.

Agree. Additional references have been included in lines 487–496, 500–501, and 536–543 as suggested.

Agree. The correction noted in lines 551–553 has been implemented.

We have also made minor revisions throughout the text to improve clarity and conciseness by rephrasing several paragraphs, as well as several contextual changes to improve the overall readability.

Additionally, we have moved Table 2 to the Supplementary Materials (now Supplementary Table 2), as it presents clinical trial data, while the main focus of the review is on genetic aspects. We believe this placement is more appropriate and consistent with the scope of the manuscript.

We sincerely appreciate the reviewer’s attention to detail, which has contributed meaningfully to enhancing the clarity and professionalism of our work.